# ADARs act as potent regulators of circular transcriptome in cancer

Haoqing Shen [1,2], Omer An [1], Xi Ren[1], Yangyang Song [1], Sze Jing Tang [1], Xin-Yu Ke[1,2], Jian Han [1], Daryl Jin Tai Tay [1], Vanessa Hui En Ng [1], Fernando Bellido Molias [1], Priyankaa Pitcheshwar [1,2], Ka Wai Leong [1], Ker-Kan Tan [3,4,5], Henry Yang [1] & Leilei Chen [1,2,5✉]

Circular RNAs (circRNAs) are produced by head-to-tail back-splicing which is mainly facilitated by base-pairing of reverse complementary matches (RCMs) in circRNA flanking introns. Adenosine deaminases acting on RNA (ADARs) are known to bind double-stranded RNAs for adenosine to inosine (A-to-I) RNA editing. Here we characterize ADARs as potent regulators of circular transcriptome by identifying over a thousand of circRNAs regulated by ADARs in a bidirectional manner through and beyond their editing function. We find that editing can stabilize or destabilize secondary structures formed between RCMs via correcting A:C mismatches to I(G)-C pairs or creating I(G).U wobble pairs, respectively. We provide experimental evidence that editing also favors the binding of RNA-binding proteins such as PTBP1 to regulate back-splicing. These ADARs-regulated circRNAs which are ubiquitously expressed in multiple types of cancers, demonstrate high functional relevance to cancer. Our findings support a hitherto unappreciated bidirectional regulation of circular transcriptome by ADARs and highlight the complexity of cross-talk in RNA processing and its contributions to tumorigenesis.

[1] Cancer Science Institute of Singapore, National University of Singapore, Singapore, Singapore. [2] Department of Anatomy, Yong Loo Lin School of Medicine, National University of Singapore, Singapore, Singapore. [3] Department of Surgery, Yong Loo Lin School of Medicine, National University of Singapore, Singapore, Singapore. [4] Division of Colorectal Surgery, University Surgical Cluster, National University Health System, Singapore, Singapore. [5] NUS Centre for Cancer Research, Yong Loo Lin School of Medicine, National University of Singapore, Singapore, Singapore. ✉email: polly_chen@nus.edu.sg

Unlike canonical linear RNAs, circular RNAs (circRNAs) are a type of RNA molecules with a covalently closed continuous loop structure. Since the circular form of RNA in the cytoplasm fraction of eukaryotic cells was first observed using electron microscope in 1979[1], circRNAs have been identified in different eukaryotes including plants, fungi, mice, and humans[2]. However, in the following decades, because of their naturality of low abundance and non-coding feature, the vast majority of circRNAs remained neglected. Only with recent advances in high-throughput sequencing, circRNAs have been characterized as ubiquitously expressed, biologically conserved and tissue-specific RNA molecules[3]. Diverse functions of circRNAs include competing with linear splicing, sponging microRNA (miRNA), interacting with RNA-binding proteins (RBPs), and producing small peptides[3]. Importantly, aberrantly expressed circRNAs have been found in many diseases such as neurological diseases, cardiovascular diseases, and cancers[4]. Since circRNAs are ubiquitous and functional, it is worth further investigation of precise mechanisms underlying the regulation of circRNA biogenesis in cells.

CircRNAs are generated by "back-splicing", which is splicing between a downstream 5′ splice donor and an upstream 3′ splice acceptor[5]. This process requires spatial proximity of non-sequential splice sites, which is usually facilitated by RBPs which bind to flanking introns[5,6] and/or base-pairing formed by reverse complementary matches (RCMs) in flanking introns such as inverted repeat Alu elements (IRAlus)[7,8]. RBPs can also facilitate or disrupt the intra-intronic base-pairing. DExH-box helicase 9 (DHX9) negatively regulates circRNA biogenesis by binding to and unwinding the base-paired IRAlus in flanking introns[9]. On the contrary, nuclear factor 90 (NF90) and its 110 kDa isoform NF110 bind to the base-pairs formed by flanking introns, leading to an increased circRNA production[10]. However, the role of other RBPs (particularly dsRNA-binding proteins) in regulating circRNA biogenesis remains largely unexplored.

Adenosine deaminases acting on RNA (ADARs) protein family, known to preferentially bind to dsRNAs formed by IRAlu elements[11], holds great potential as a potent circRNA regulator. Upon dsRNA binding, ADARs may catalyze adenosine to inosine (A-to-I) editing, the most prevalent type of RNA editing in eukaryotes[12], in their bound dsRNAs. Till now, a few studies have reported controversial findings about the effect of ADARs on circRNA biogenesis. It has been suggested that ADARs could suppress the generation of circRNAs by editing and "melting" the dsRNA[8,13]. However, another study claimed that ADARs alone had no major effect on circRNA biogenesis, although double knockdown of ADAR1 and DHX9 repressed circRNA biogenesis to a greater extent than the single knockdown of each gene[9]. Besides, there is still a lack of experimental evidence supporting that ADARs-mediated editing can destabilize (and unwind) dsRNAs formed by IRAlu elements. The role of ADARs in circRNA biogenesis warrants a deeper investigation from the facts that: (1) ADARs preferentially edit A:C mismatches rather than A-U base pairs[14], presumably resulting in a more stable secondary structure; (2) through editing, ADARs can strengthen or weaken binding of RBPs by altering RNA sequences of *cis*-elements and/or creating or destroying RBP binding motifs[15–17]; and (3) independent of their editing function, ADARs can also block the access of other RBPs (e.g. U2AF65) to the latter's original binding sites, contributing to changes in splicing[15]. These abovementioned facts challenge the common opinion that ADARs function as repressor of circRNA biogenesis, presumably dependent on their editing capability.

It is known that the differentially expressed ADARs and its resultant dysregulation of A-to-I RNA editome are implicated in multiple cancer types such as esophageal squamous cell carcinoma (ESCC), hepatocellular carcinoma (HCC), colorectal cancer (CRC), breast cancer, and gastric cancer[18–24]. Herein, we comprehensively define the regulatory role of ADARs in circRNAs and explore the functional relevance of target circRNAs to cancer. We uncover over a thousand circRNAs either promoted or repressed by ADAR1 and/or ADAR2 via editing-dependent or -independent mechanisms. Next, our mechanistic investigation deciphers editing-dependent mechanisms of action by which editing can stabilize or destabilize secondary structures formed between RCMs within the flanking introns via correcting A:C mismatches to I(G)-C pairs or creating I(G)·U wobble pairs, respectively. We also find that editing facilitates the recruitment of RBPs such as PTBP1 to regulate back-splicing. Moreover, we show that these ADARs-regulated circRNAs (ARcircs) are not merely by-products of back-splicing, but indeed influence tumorigenesis. Our findings provide a previously undescribed bidirectional regulation of circular transcriptome by ADARs and highlight a complex crosstalk between RNA editing machinery and circRNA biogenesis and its implications in cancer.

## Results

**ADARs regulate circRNA biogenesis bidirectionally**. To query the role of ADARs in regulating circRNAs, we modulated the expression level of ADAR1 or ADAR2 (ADAR1/2) by either forced expression or silencing in EC109 which is an esophageal squamous carcinoma cell line that has been frequently used for ADARs and A-to-I editing research[23–25]. Untreated or RNase R-treated RNA samples were subsequently sent for total RNA sequencing (RNA-Seq) or circRNA sequencing (circRNA-Seq), respectively (Fig. 1a). We applied an in-house pipeline for circRNA detection and identified a total of 37,916 circRNAs. To ensure the reliability of our analysis, we compared the performance for circRNA identification between our pipeline and two commonly used benchmark methods CIRI2 and CIRCexplorer2[26–28] and obtained a high percentage of overlapping circRNAs between our in-house pipeline and CIRI2 or CIRCexplorer2 (87% for CIRCexplorer2; 70% for CIRI2) (Supplementary Fig. 1a). With our stringent filter criteria (see Methods), a total of 650 and 868 circRNAs were identified as high-confidence ADAR1 or ADAR2-regulated circRNAs (ARcircs), respectively (Fig. 1b and Supplementary Data 1). Intriguingly, both ADAR1 and ADAR2 proteins were found to regulate circRNAs in both directions (Fig. 1b, c). Unlike ADAR1 which exerts its suppressive or promoting effect on approximately the same amount of circRNAs (promoting: 48% *vs* repressing: 52%), ADAR2 is most likely to be a potent repressor of circRNAs rather than an enhancer (promoting: 15% *vs* repressing: 85%) (Fig. 1b, c). Among approximately a hundred circRNAs regulated by both ADAR proteins (defined as common circRNAs), no circRNA was found to be regulated by ADAR1/2 in opposite directions (Fig. 1c). Of note, using the same filter criteria, 93% (1,313/1,406) of ARcircs identified by our pipeline were also found by CIRCexplorer2 and demonstrated the same pattern of changes upon modulation of ADAR1/2 expression (Supplementary Fig. 1b and Supplementary Data 1). To rule out the possibility that such effects might arise from the changes in linear mRNA expression, we analyzed expression changes in all detected circRNAs and their host gene transcripts upon modulation of ADAR1/2 expression and observed drastic expression changes in circRNAs, but not their corresponding linear mRNAs (Fig. 1d). Our finding is consistent with previous reports that ADARs has no major effect on global gene expression, even those undergoing A-to-I editing[29–31].

To confirm these findings, we randomly selected 21 candidate circRNAs for experimental validation (Supplementary Fig. 1c). Upon RNase R digestion, all linear forms underwent more than

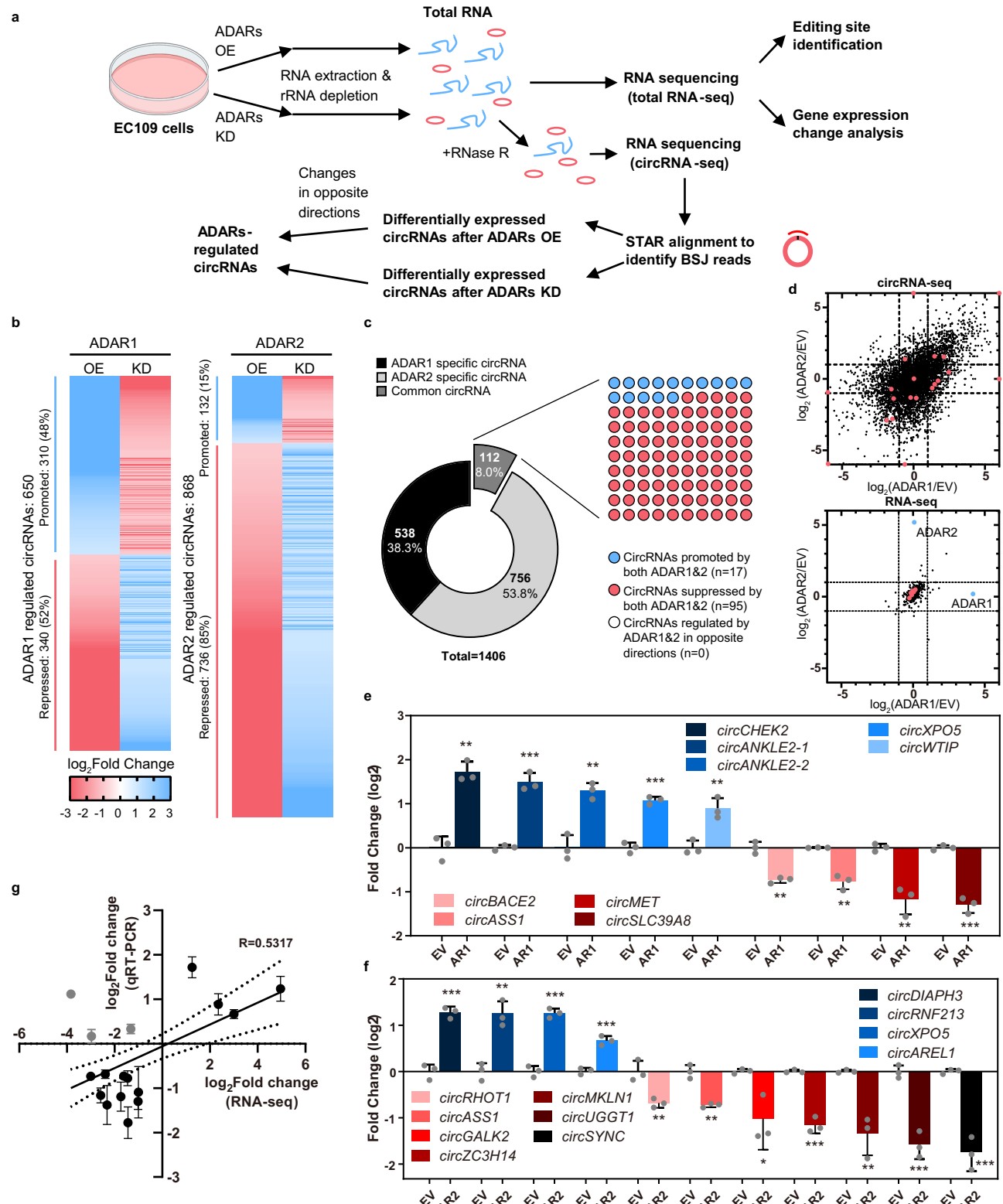

50-fold reduction in their expression; while circRNAs demonstrated strong resistance to the digestion, indicating that these candidate circRNAs are truly circularized RNA molecules (Supplementary Fig. 1d). Sanger sequencing analyses of purified circRNA products further confirmed these back-splicing events (Supplementary Fig. 1e). We then went on to validate the regulatory effects of ADARs on these candidate ARcircs identified by circRNA-Seq. Expression change was successfully verified for

20 out of 23 candidate ARcircs (ADAR1-regulated: 9 out of 10; and ADAR2-regulated:11 out of 13), but not for their host linear mRNAs (Fig. 1e–g and Supplementary Fig. 1f). Of note, 2 ARcircs circASS1 and circXPO5, were confirmed to be common targets for both ADARs (Fig. 1e–g and Supplementary Fig. 1e). Moreover, these ARcircs were further validated in ADAR1/2-knockdown cells (Supplementary Fig. 1g, h). These data indicate that ADARs indeed function as potent bidirectional regulators of circular

**Fig. 1 ADAR1 and ADAR2 regulate circRNA biogenesis bidirectionally. a** Workflow for identification of ADARs-regulated circRNAs (ARcircs). **b** Heat maps indicating the fold change in expression of candidate circRNAs, upon modulation of ADAR1 or ADAR2 expression through lentivirus-based knockdown (KD) or overexpression (OE). A relative decrease in the KD or OE samples is indicated as blue, while an increase is indicated as red. **c** Doughnut chart depicts the percentage of circRNAs regulated specifically by either ADAR1 or ADAR2, or by both ADAR proteins. 10 × 10 dot plot illustrates the number of common circRNAs which are regulated by ADAR1 and ADAR2 in either same or opposite direction. **d** Scatterplots displaying the fold changes in expression levels of all 37,916 detected circRNAs (upper) or their corresponding host genes (lower), upon overexpression of ADAR1/2 versus empty vector control in EC109 cells. A total of 20 randomly selected circRNAs from (**b**) and their corresponding host genes are indicated as red dots. Blue dots indicate ADAR1 and ADAR2. **e, f** Quantitative real-time PCR (qRT-PCR) validation of the indicated circRNAs. **g** Correlation between fold change calculated from circRNA-Seq and qRT-PCR validation data. Dash lines show 95% confidence interval. CircRNAs not validated are showed in grey color. **e–g** Data are presented as the mean ± S.D. of technical triplicates from a representative experiment of 2 independent experiments (unpaired, two-tailed Student's $t$-test; *$P < 0.05$; **$P < 0.01$; ***$P < 0.001$). Exact P values and source data are provided in Source Data file.

transcriptome, with ADAR2 appearing skewed towards a repressor. For those circRNAs regulated by both ADARs, they are most likely to be regulated in the same direction.

**ADARs regulate circRNAs through or beyond their editing function.** Base-pairing of reverse complementary matches (RCMs) residing in circRNA flanking introns facilitates circRNA production[7,8]. To further dissect the mechanism underpinning the regulatory role of ADARs in circRNA biogenesis, we first identified 41,551 high-confidence A-to-I editing sites from the total RNA-Seq data and 1,043 ARcircs with ≥1 RCM locating in their flanking introns from the circRNA-Seq data (Fig. 2a and Supplementary Data 2). To ensure the specificity of identified RNA editing sites, we checked the proportion of each possible type of mismatches and found that A to G accounts for approximately 90% of all detected mismatches, consistent with previous studies reporting A-to-I editing as the most common type of RNA editing in humans[32,33] (Supplementary Fig. 2a). Moreover, we analyzed the sequence preference for neighboring nucleotides surrounding editing sites (± 2 nt) and found "G" is preferred to be excluded at 5′ neighbour but included at 3′ neighbour of editing sites, as reported previously[34,35] (Supplementary Fig. 2b). We then went on to analyze the distribution of these editing sites and RCMs across the flanking introns (Fig. 2a). Not surprisingly, RCMs are obviously enriched in the intronic region proximal to the back-splicing junctions (−500 nt ~ +500 nt; black dots, Fig. 2b), indicating that intronic matches between flanking introns are truly involved in back-splicing. A previous study reported that both RCMs and editing sites from RADAR (Rigorously Annotated Database of A-to-I RNA editing)[36] are preferentially distributed near the splice sites of circularized exons[8]. However, from our analysis, the locations of editing sites identified in either our own EC109 total RNA-seq data or the RADAR database are not enriched in the proximal back-splice junction (BSJ) region (Fig. 2b, red and blue lines). A recent study also suggested that such an enrichment was not observed in the flanking introns of circRNAs regulated by ADARs in the mouse bone marrow or liver tissue samples[37].

We next questioned if the editing capability of ADARs is indispensable for their regulation of circRNAs. To this end, the ADAR1/2 mutants depleted of either editing activity only (DeAD mutants)[38] or both RNA binding and editing capabilities (EAA mutants)[39,40] were generated. Upon overexpression of each wildtype or mutant form, the ADAR1/2 EAA mutant was incapable of regulating all ARcircs (Fig. 2c, d and Supplementary Fig. 2c, d), suggesting that RNA binding ability is critical for ADARs to regulate ARcircs; unlike the EAA mutant, the DeAD mutant was able to modulate the expression of approximately half of ARcircs such as *circXPO5*, *circASS1*, and *circRNF213*, to a similar or less extent than the wildtype form (Fig. 2c, d and Supplementary Fig. 2c, d). These data suggested that ADARs can regulate circRNAs through their editing-dependent and/or

independent functions. To further interrogate whether such an editing-dependent/independent regulation of circRNA biogenesis can be observed in a transcriptome-wide manner, we over-expressed the ADAR1/2 DeAD mutant or the empty vector (EV) control in EC109 cells and performed circRNA-Seq to identify editing-dependent and -independent ARcircs. From this batch of circRNA-Seq, we could detect 76.3% (1,073/1,406) of ARcircs identified from our previous circRNA-Seq. Among these 1,073 ARcircs, 767 were identified as editing-dependent ARcircs and 306 as editing-independent ones regulated by ADAR1 and/or ADAR2 (Methods and Supplementary Data 3). All these findings strongly indicate that ADARs can bidirectionally regulate circRNAs via editing-dependent and/or independent mechanisms in a transcriptome-wide manner.

**ADAR1 promotes *circCHEK2* biogenesis via its direct binding and editing of *circCHEK2* flanking introns.** So far, there is a lack of experimental evidence about the mechanism underpinning the regulation of circRNA biogenesis via ADAR-mediated editing. *CircCHEK2*, an editing-dependent ARcirc generated by back-splicing between exon 3 and 9 of its host gene *CHEK2* (Fig. 3a), was chosen as an exemplary target for further study. We first analyzed the publicly available ADAR1 RNA immunoprecipitation sequencing (fRIP-Seq) dataset[41,42] and found that ADAR1 binding peaks enriched in both flanking introns 2 and 9, especially the identified RCM pair with the highest BLAST score (Fig. 3a). A high probability of dsRNA formation between the predicted RCM was supported by secondary structure prediction using RNAfold[43]. Further, by performing RNA immunoprecipitation (RIP) assay, we confirmed the association of ADAR1 with the dsRNA structure formed between the identified RCMs in vivo (Fig. 3b). We then provided experimental evidence that upon ADAR1 overexpression, three editing sites (sites #1, #2 and #3) within RCMs could be detected with editing frequencies ranging from 19% to 28.6% (Fig. 3c). Moreover, the site #1 is located in a previously reported ADAR1 binding motif[44] (Fig. 3c). All these findings suggest that ADAR1 indeed binds and edits the dsRNA structure formed between RCMs located in the flanking introns of *circCHEK2*.

To explore whether editing of RCM has an effect on *circCHEK2* expression, we generated a *circCHEK2* minigene containing the partial sequence of *CHEK2* gene, including the entire exons 3-9, part of exon 2, exon 10 and flanking introns 2 and 9 (Fig. 3d). We transfected the *circCHKE2* minigene together with the wildtype ADAR1 or DeAD mutant into cells. Like endogenous *circCHEK2*, exogenous *circCHEK2* derived from the minigene was also regulated by ADAR1 dependent on the latter's editing function (Fig. 3e). Increased editing was observed at the same editing sites within the flanking intronic sequence of the exogenous pre-mRNA (Fig. 3c). It is known that cellular machineries recognize inosine as guanosine (G), due to their high structural similarity. To further understand whether all three

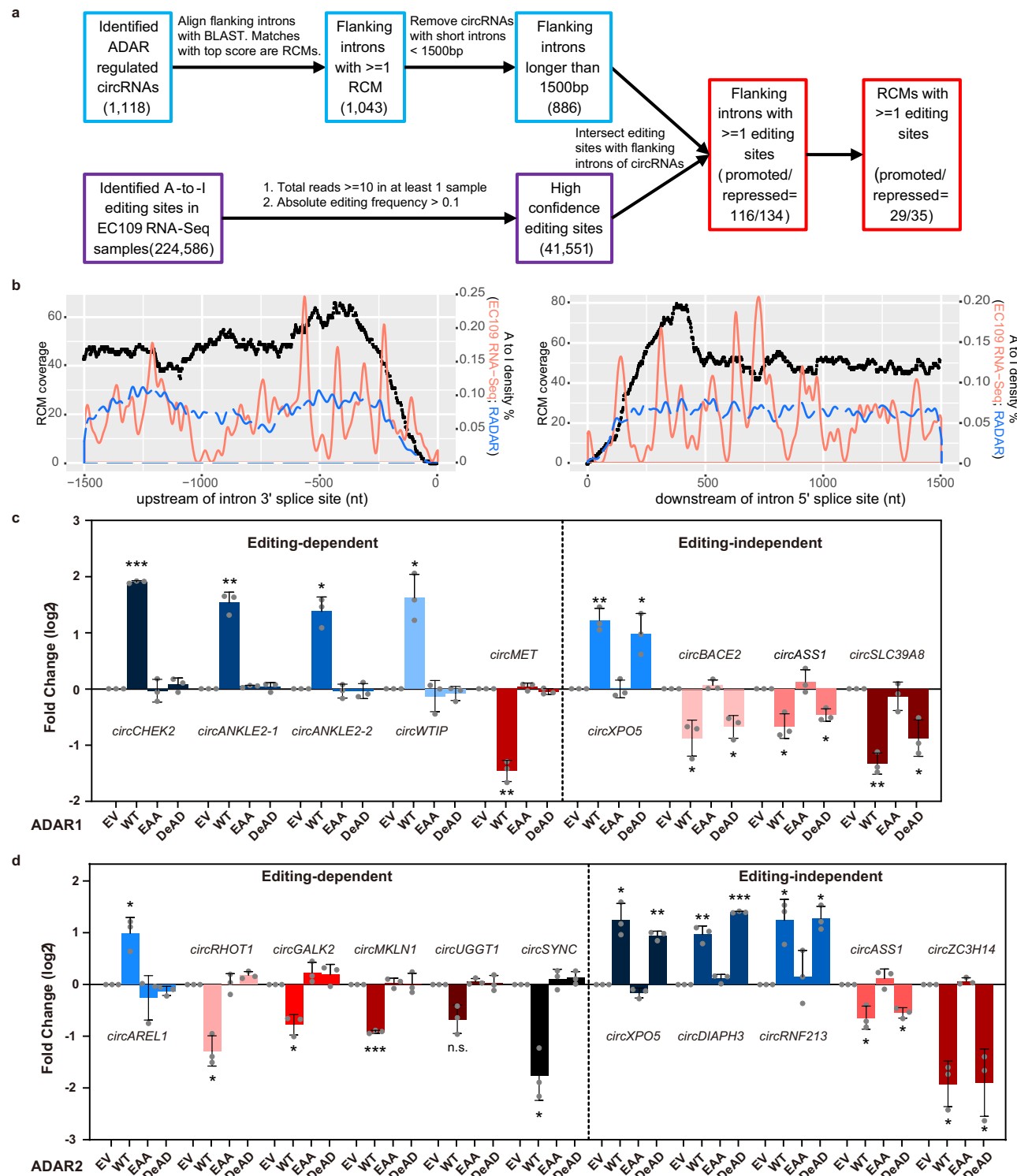

**Fig. 2 ADAR1/2 regulate circRNA biogenesis via either editing-dependent or -independent mechanisms. a** Workflow for identification of reverse complementary matches (RCMs) and A-to-I editing sites at flanking introns of ARcircs. **b** Distribution of RCMs (left Y-axis) and editing sites (right Y-axis) across the flanking intronic region spanning 1,500nt upstream (left panel) and 1,500nt downstream (right panel) of back-splicing junction site of ARcircs. Black dotted lines indicate the distribution of RCMs. Red or blue lines indicate the distribution of editing sites identified from our EC109 RNA-seq data or the RADAR database, respectively. **c**, **d** qRT-PCR analysis of expression change of the indicated ARcircs, upon overexpression of the wildtype (WT), EAA mutant, or DeAD mutant form of ADAR1 (**c**) or ADAR2 (**d**) versus empty vector (EV) control in EC109 cells. Each dot represents the mean value of technical triplicates from an independent experiment. Data are presented as the mean ± S.D. of 3 biological replicates (paired, two-tailed Student's *t*-test; *P < 0.05; **P < 0.01; ***P < 0.001). Exact P values and source data are provided in Source Data file.

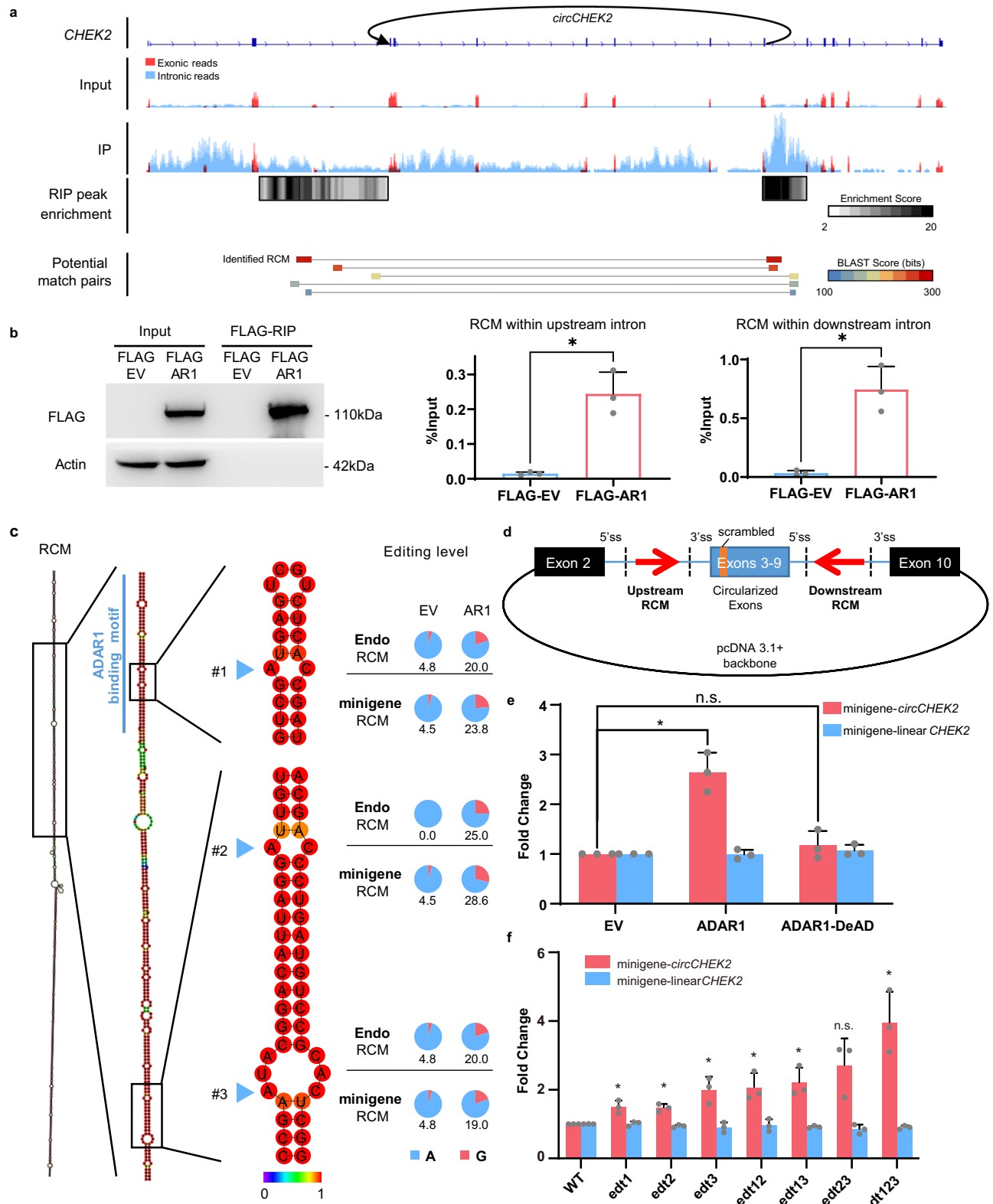

editing sites are involved in such regulation, we introduced an A-to-G mutation into the minigene at each editing site to mimic a fully edited site. Intriguingly, each single, double, or triple combination of these mutations led to increased expression of minigene-*circCHEK2* (Fig. 3f). Of note, these three editing sites demonstrated synergistic effect on promoting *circCHEK2* biogenesis (Fig. 3f). Collectively, ADAR1-mediated editing of RCMs can promote *circCHEK2* biogenesis.

**A-to-I editing may alter circRNA production via stabilizing or destabilizing dsRNA formed between RCMs**. As dsRNA between circRNA flanking introns is one of the key factors for circRNA biogenesis, we next asked whether ADAR1-mediated editing of RCMs alters the secondary structure. Based on the in silico secondary structure prediction, all three edited adenosines form A:C mismatches in the dsRNA, and editing at A:C mismatches which changes A:C to I(G)-C may enable a more perfect

**Fig. 3 ADAR1 binds and edits *circCHEK2* RCM to promote *circCHEK2* biogenesis. a** Genome browser tracks of *CHEK2* loci reveal ADAR1 binding peaks (top) from ADAR1-fRIPseq data and predicted RCM pairs within *circCHEK2* flanking introns (bottom). Black arrow: the circular junction site of *circCHEK2* in a 5′-3′ direction. Reads mapped to exonic or intronic regions (GENCODE annotation) are colored in red or blue, respectively. Potential match pairs are indicated in different colors and the pair with the highest BLAST score is defined as "Identified RCM". **b** RIP-qPCR analysis of the association of ADAR1 protein to the *circCHEK2* RCM region in EC109 cells transfected with FLAG empty vector (FLAG EV) or FLAG-ADAR1 (FLAG AR1). WB and qPCR analyses of FLAG-RIP immunoprecipitates are shown in the left and right panels, respectively. **c** Secondary structure formed by RCMs of *circCHEK2*, as predicted by RNAfold (left). Location of a reported ADAR1 binding motif is indicated by blue line. Blue arrows indicate 3 editing sites identified within RCMs of *circCHEK2*. Base-pair probabilities are shown by a color spectrum (middle). Pie charts illustrating the editing frequency (indicated by red slice) of each editing site in the indicated samples (right). Editing frequency of each editing site was measured using TA cloning (see Methods). **d** Schematic diagram illustrating the structure of *circCHEK2* minigene. A 20-bp sequence of exon 3 was scrambled to distinguish minigene-produced transcripts from endogenous transcripts. **e** Fold change in expression of minigene-produced *circCHEK2* and linear *CHEK2*, upon overexpression of WT or DeAD ADAR1, compared to EV control. **f** Fold change in expression between *circCHEK2* and linear *CHEK2* derived from the WT or mutated minigenes carrying A-to-G mutation(s) at editing sites. Edt1, A-to-G mutation at site #1; edt12, A-to-G mutations at sites #1 and #2, and so forth. **b, e, f** Data are presented as the mean ± S.D. of 3 biological replicates. Each dot represents the mean value of technical triplicates from an independent experiment. Data is presented as mean ± S.D. of 3 biological replicates. Statistical significance is determined by paired, two-tailed Student's *t*-test (*, $P < 0.05$; n.s., not significant). Exact $P$ values and source data are provided in Source Data file.

secondary structure, which potentially facilitates circRNA production (Fig. 4a). It has been known that tightly folded RNAs travel more rapidly than unfolded RNAs of the same length or molecular weight[45]. To test our hypothesis, we generated RNA probes containing the *circCHEK2* RCM sequence with or without single, double, or triple A to G mutations at the three editing sites and performed native polyacrylamide gel electrophoresis (PAGE). As expected, probes with mutations at all three editing sites (edt123) migrated more rapidly on gel (Fig. 4b), suggesting that editing may enable RCMs within the flanking intronic sequence to form a more compact structure to stabilize the dsRNA.

To obtain more experimental evidence supporting that editing of RCMs can alter dsRNA structure, we selected 6 additional editing-dependent ARcircs identified by circRNA-Seq (Supplementary Data 3), including 3 ADAR1/2-promoted circRNAs (*circASH1L*, *circANKLE2-1* and *circRNF114*) and 3 ADAR1/2-repressed circRNAs (*circSYNC*, *circDHX34* and *circRHOT1*). From our RNA-Seq data, all 6 ARcircs have editing sites within their RCMs demonstrating ≥ 10% increase in editing frequency upon overexpression of the corresponding ADAR protein. Using the same strategy, we found that the majority of editing sites within RCMs of *circASH1L*, *circANKLE2-1* and *circRNF114* locate at A:C mismatches where editing was predicted to lead to more compact dsRNA structures (Fig. 4c); on the contrary, upon editing, RCMs of *circSYNC*, *circDHX34* and *circRHOT1* hypothetically form looser dsRNA structures via changing A-U base pairs to weaker G.U wobble base pairs or affecting the structures of neighboring regions (Fig. 4d). Intriguingly, native PAGE analysis showed that for *circASH1L*, *circANKLE2-1* and *circRNF114*, the edited RCM probes with A-to-G mutations at all editing sites (indicated by arrows, Fig. 4c, d) migrated faster in the gel than the unedited/wildtype probes (Fig. 4e, left panel); while for *circSYNC*, *circDHX34* and *circRHOT1*, the edited probes migrated slightly slower than the unedited/wildtype probes (Fig. 4e, right panel). Therefore, there could be a universal editing-dependent mechanism by which ADARs regulate circRNA biogenesis via editing-mediated change in the secondary structure formed by flanking introns.

**A-to-I editing enhances PTBP1 binding to flanking introns of *circCHEK2* to promote its biogenesis.** Other than causing structural changes, A-to-I editing of intronic sequence has been proved to be a regulator of splicing by creating or modifying auxiliary *cis*-acting elements for splicing factor binding[15,17]. Since the canonical machinery of spliceosome also functions in circRNA biogenesis, we next asked whether editing could facilitate

the binding of splicing regulators and affect back-splicing via changing *cis*-acting elements. To this end, we predicted RBPs which demonstrate binding preference near *circCHEK2* editing sites using RBPmap[46]. Two RBPs, TDP43 and PTBP1 with a respective binding motif near the editing site #1 and #2, are of particular interest (Fig. 5a). RNA pulldown assay was performed by incubating the whole cell lysates with the wildtype (WT) or triple mutant (edt123) RNA probe. Intriguingly, PTBP1 was found to bind more strongly to the edt123 than the WT probe, while TDP43 did not show any distinct binding preference between 2 probes (Fig. 5b and Supplementary Fig. 3a). We further performed PTBP1 RIP assay in EC109 cells with or without overexpression of ADAR1 and found that binding of PTBP1 to the *circCHEK2* RCM region was significantly enhanced upon overexpression of ADAR1 (Fig. 5c). Intriguingly, the proportion of edited RCM transcripts (shown by editing frequencies of all 3 editing sites) was increased in PTBP1 RIP products when compared to the 'Input' samples, particularly in ADAR1-overexpressing cells (Fig. 5d), further confirming that editing enhances PTBP1 binding to the *circCHEK2* RCM region.

Next, we determined the regulatory effect of PTBP1 and TDP43 on *circCHEK2* biogenesis. In the absence of PTBP1 knockdown, there was an approximately 5-fold higher expression of *circCHEK2* derived from the triple mutant (edt123) minigene than the WT counterpart; however, upon knockdown of PTBP1, the difference in the efficiency of *circCHEK2* production between edt123 and WT minigene was significantly attenuated (Fig. 5e and Supplementary Fig. 3b). However, such changes were not observed upon silencing of TDP43 (Fig. 5f and Supplementary Fig. 3c). Endogenously, silencing of PTBP1 also reduced the promoting effect on *circCHEK2* biogenesis caused by ADAR1 overexpression (Fig. 5g and Supplementary Fig. 3d). Previous study reported that PTBP1 could affect the translation of ADAR1 in glioma cells[47], which may serve as an additional regulatory mechanism of PTBP1 on *circCHEK2*. However, we did not observe any obvious reduction in ADAR1 protein level upon silencing of PTBP1 (Supplementary Fig. 3e). All these data suggested that besides editing-mediated change in the secondary structure formed by *circCHEK2* flanking introns, editing can also enhance PTBP1 binding to the flanking introns and promote *circCHEK2* biogenesis.

**Editing can alter RBP binding sites in the flanking introns of circRNAs in a transcriptome-wide manner.** Inspired by our observations, we next sought to investigate whether editing-mediated changes in binding sites of RBPs may serve as a general mechanism to regulate circRNA biogenesis. With the same

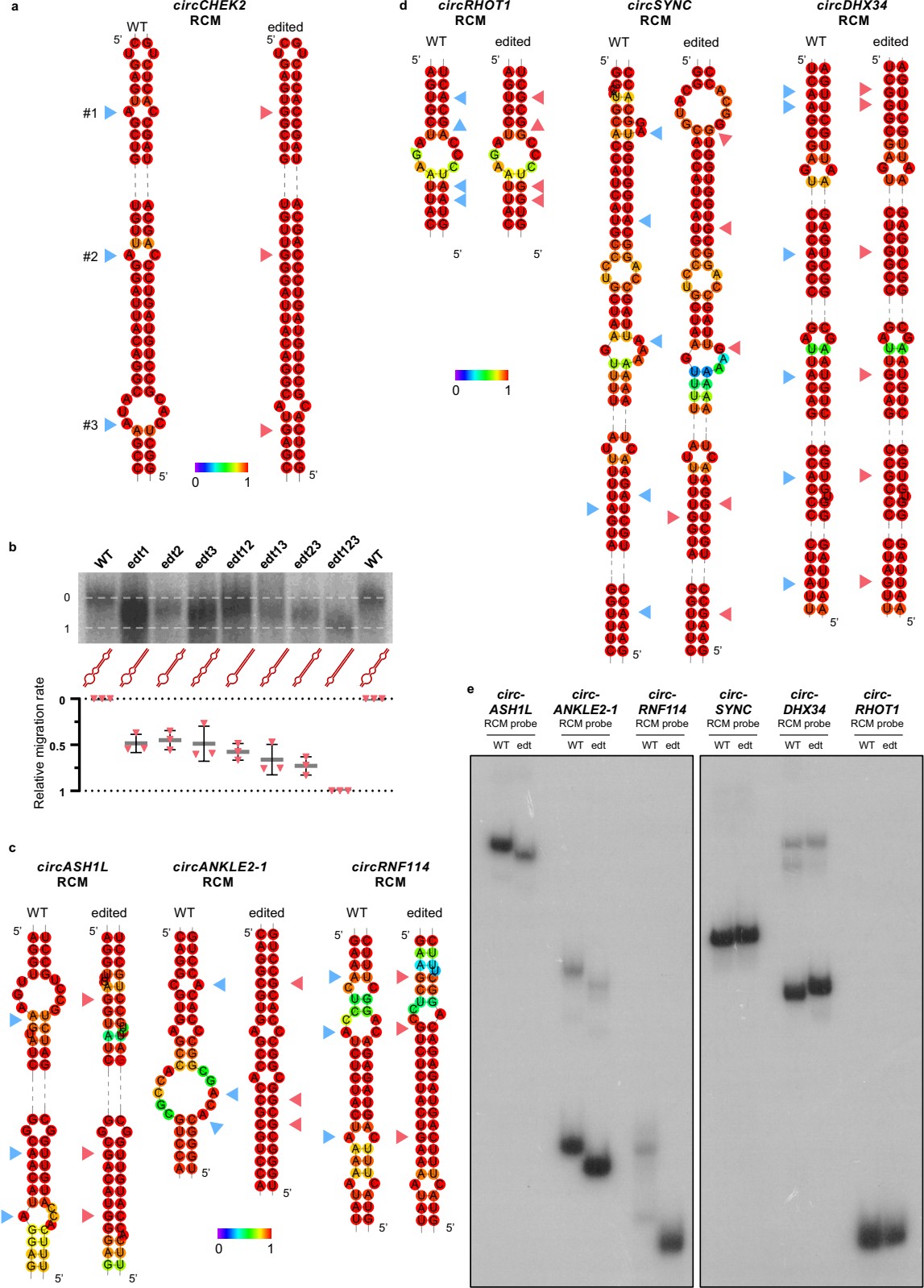

stringent filter (Fig. 2a), we identified 571 editing sites distributed within flanking introns of 92 editing-dependent ARcircs. We next retrieved the sequence surrounding editing sites (±10 nt) and analysed RBP binding motifs before and after editing using RBPmap[46], followed by the calculation of the number of circRNAs which have altered RBP binding sites on flanking introns due to editing. We found that among 132 analysed RBPs with annotated binding sites in RBPmap, 129 RBPs, including PTBP1 and those which have been shown to regulate circRNA biogenesis such as MBNL1[5], FUS[48], SFPQ[49], HNRNPL[50], KHSRP[50], and QKI[6], were found to have editing-mediated changes in their binding sites at flanking introns of more than 10 editing-dependent ARcircs (Fig. 5h and Supplementary Data 4), implying that altering RBP binding affinity is an important mechanism for

**Fig. 4 A-to-I editing alters the dsRNA structure formed by RCMs within flanking introns of circRNAs. a** Predicted secondary structures formed by *circCHEK2* RCM with or without A-to-I(G) editing by RNAfold. Partial RNA structures which contain editing sites and neighboring sequences are shown. Blue and red arrows indicate unedited/wildtype (WT) adenosines and edited/mutated (edited) sites, respectively. **b** Migration on native polyacrylamide gel of RNA probes containing the *circCHEK2* RCM sequence with or without A-to-I(G) editing at each editing site. Simplified secondary structure of each probe is shown. Calculation of the relative migration rate was discussed in Methods. Data are presented as the mean ± S.D. of biological triplicates. **c, d** Predicted secondary structures formed by RCMs of ADAR1/2-promoted circRNAs (*circASH1L*, *circANKLE2-1* and *circRNF114*) **c** and ADARs-repressed circRNAs (*circRHOT1*, *circSYNC* and *circDHX34*) **d** by RNAfold, with or without editing. Partial RNA structures which contain editing sites and neighboring sequences are shown. Blue and red arrows indicate WT adenosines and edited sites, respectively. **e** Migration on native polyacrylamide gel of RNA probes containing the wildtype (WT) and edited (edt) partial RCM sequences of the indicated circRNAs. Representative result of *n* = 2. **a, c, d** Base-pair probabilities are shown by a color spectrum. Source data are provided in Source Data file.

editing to regulate circRNA biogenesis. Taken together, editing can not only alter the stability of secondary structure formed between RCMs, but also affect RBP binding to flanking intronic sequences, leading to changes in circRNA production.

**ADARs-mediated circRNA regulation exists in multiple cancer types**. ADAR1 and ADAR2 are ubiquitously expressed in many tissue types[11]. We wondered if ADARs function as potent regulators of circular transcriptome in multiple cancer types. To address this, we selected five validated ARcircs and detected their expression changes upon overexpression of the wildtype or mutant form of ADAR1/2 in MB231 (breast cancer cell line), MKN28 (gastric tubular adenocarcinoma cell line), SNU398 (hepatocellular carcinoma cell line), and HCT15 (colorectal cancer cell line). Intriguingly, we observed the same pattern of editing-dependent or independent regulation of ARcircs in these cell lines as EC109 cells (Fig. 6a–d and Supplementary Fig. 4a, b). We then investigated the expression pattern of *circCHEK2* and the association between expression levels of *ADAR1* and *circCHEK2* in 17 matched pairs of primary HCC and non-tumor (NT) liver samples as well as 20 matched pairs of primary colorectal cancer (CRC) and NT colon samples. We found that 41% (7 out of 17) and 60% (12 out of 20) of HCC and CRC patients demonstrated a ≥ 2-fold increase in *circCHEK2* expression in tumors compared to their NT samples, respectively (Fig. 6e, f, upper panels). Next, both HCC and CRC patients were stratified into two groups: ADAR1-down and ADAR1-up, based on the decreased or increased expression of ADAR1 in tumors compared to their matched NT samples, respectively (Fig. 6e, f, lower panels). We found that in the ADAR1-down or ADAR1-up group of HCC patients, 3 out of 6 (50%) or 6 out of 11 (54.5%) showed ≥ 2-fold decrease or increase in *circCHEK2* expression in tumors, respectively (Fig. 6e). Likewise, in the ADAR1-down or ADAR1-up group of CRC patients, 4 out of 6 (67%) or 8 out of 14 (57%) showed ≥ 2-fold decrease or increase in *circCHEK2* expression in tumors, respectively (Fig. 6f). These findings suggested that ADARs-mediated circRNA regulation is most likely present in multiple cancer types.

**Impacts of ARcircs on tumorigenesis**. To investigate the potential involvement of ARcircs in tumorigenesis, we utilized CasRX (also known as RfxCas13d) system[51,52] and designed guide RNAs (gRNAs) against the back-splicing junction sequence of each ARcirc for a specific and efficient knockdown without affecting their host genes expression (Fig. 7a). In 2 different types of cancer cell lines EC109 and SNU398, knockdown of *circCHEK2*, *circGALK2*, and *circSLC39A8* significantly reduced the tumorigenic ability of cells, as manifested by decreased frequencies of focus formation and colony formation in soft agar, suggesting that these ARcircs have a cancer-promoting role (Fig. 7b–e). We further provided in vivo evidence that *circCHEK2* knockdown in EC109 and SNU398 cells led to a significant

reduction in tumor growth rate than the control counterparts (Fig. 7f, g). All these data suggested that these ARcircs are of functional relevance to multiple types of cancers.

**Discussion**

Although several previous studies reported that RNA editing enzymes ADARs function as repressors of circRNA biogenesis[8,13] or have no major regulatory effect on circRNAs[9], our study demonstrates that ADARs are potent regulators of circular transcriptome and they can regulate over a thousand of circRNAs in both directions through and beyond their editing functions. However, it remains unknown what mechanisms determine the direction of circRNA regulation by ADARs. One key factor is the position of the edited adenosine within a dsRNA. Previously proposed model suggested that ADARs destabilize the secondary structure through altering A-U base pairs located in the dsRNA stem, leading to repression of circRNA biogenesis. Here, we provided experimental evidence that adenosines at A:C mismatches, which gain editing preference than those at A-U pairs[14], can stabilize the dsRNA structure formed between flanking introns, promoting circRNA biogenesis. It has been known that canonical splicing signals and spliceosomal machinery are required for back-splicing, and editing at *cis*-acting elements (e.g., branch point site, splicing enhancers/silencers) can result in changes of splicing pattern[15–17]. Therefore, the location of editing sites within the host gene transcript may influence circRNA expression. In this work, we demonstrate that ADAR1-mediated A-to-I editing can enhance binding of splicing factor PTBP1 to the flanking intron of *circCHEK2*, rendering increased expression of *circCHEK2*. Although PTBP1 is well documented as a pyrimidine-rich sequence binding protein, a previous study also showed that guanosine containing triplets contribute to PTBP1 binding[53]. This explains our observation that A-to-I (G) substitutions within *circCHEK2* RCMs could enhance PTBP1 binding. Moreover, we provided a large-scale prediction of editing-mediated changes on RBP binding motifs on flanking introns of ARcircs and found that upon editing, most analyzed RBPs have altered binding sites in flanking introns of more than 10 editing-dependent ARcircs, further suggesting that editing may regulate circRNA biogenesis through affecting RBP binding in a transcriptome-wide manner. One should note that there may be other editing-dependent and -independent mechanisms underpinning the regulation of circRNAs by ADARs, such as altering the circRNA turnover and splicing.

RNA editing, alternative splicing, polyadenylation, and back-splicing are crucial RNA processing steps that expand transcriptome diversity. As each step heavily involves base-pairing (e.g., dsRNA formation of IRAlu elements), it is not surprising that these processes undergo extensive crosstalk. These dsRNAs recruit RBPs, dramatically increasing the complexity of RNA processing network. One example is DHX9, which regulates RNA editing as a binding partner of ADARs and also suppresses circRNA biogenesis via unwinding dsRNAs formed by IRAlu

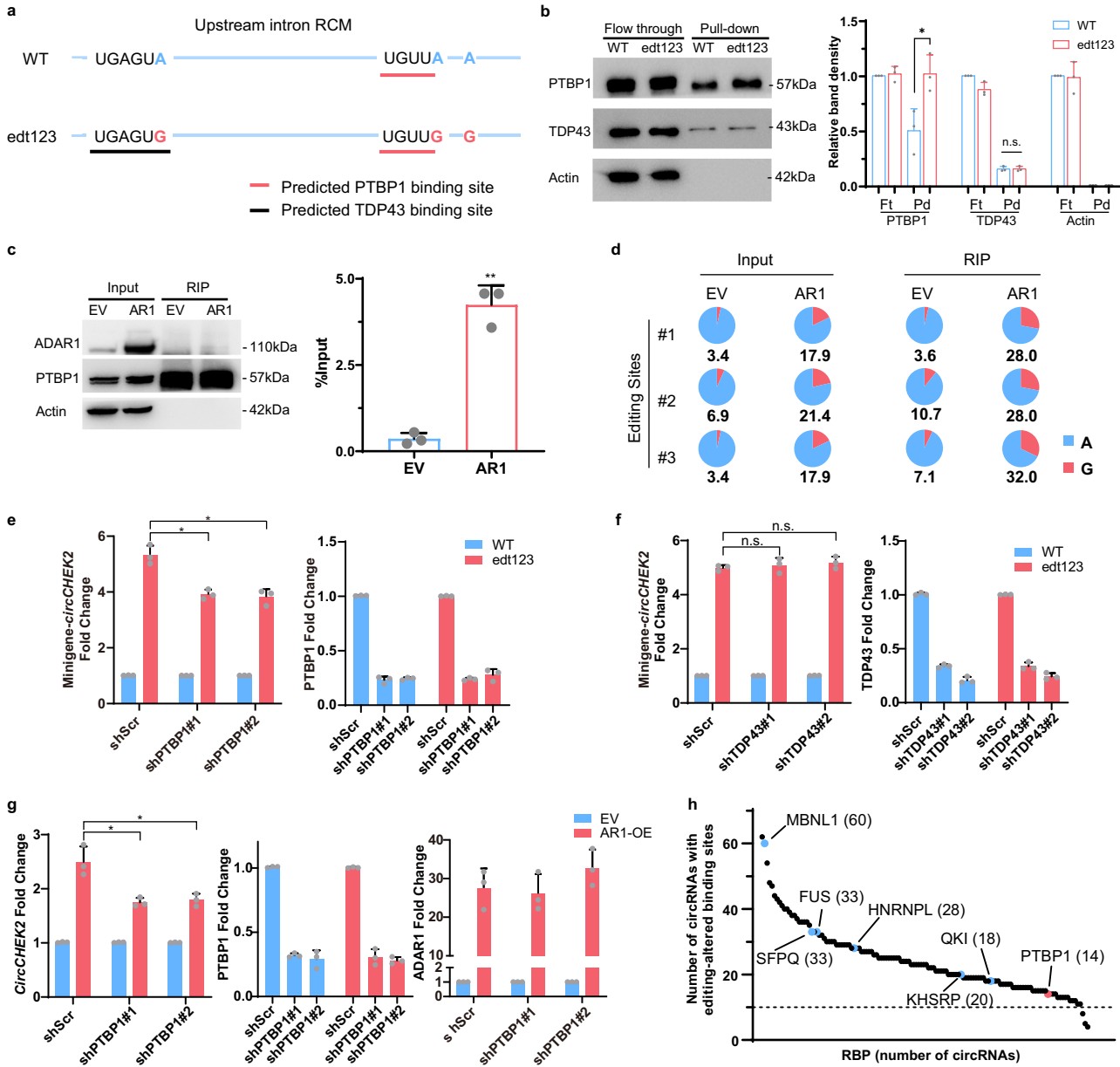

**Fig. 5 A-to-I editing enhances PTBP1 binding to intron and promote *circCHEK2* biogenesis. a** In silico prediction of PTBP1 (red line) and TDP43 (black line) binding to unedited (WT) or triple mutated (edt123) RCM sequence of *circCHEK2* using RBPmap. **b** WB analysis of RNA pull-down products showing the binding affinity of PTBP1 and TDP43 to the WT or edt123 RNA probes. **c** RIP-qPCR analysis of the binding of PTBP1 protein to the *circCHEK2* RCM region in EC109 cells transfected with ADAR1 or empty vector control (EV). WB and qPCR analyses of PTBP1 RIP immunoprecipitates are shown in the left and right panels, respectively. **d** Pie charts illustrating the editing frequency (indicated by red slice) of each editing site (#1, #2, #3) in the indicated Input or RIP samples. Editing frequency of each editing site was measured using TA cloning (see Methods). **e, f** Left panels: Fold change in expression of *circCHEK2* produced by minigenes with or without A-to-G mutations at three editing sties, upon knockdown of PTBP1 **c** or TDP43 **d**. Right panels: qPCR analysis showing the knockdown efficiency of PTBP1 and TDP43. **g** Left panel: Fold change in expression of endogenous *circCHEK2* with or without lentivirus-mediated overexpression of ADAR1 in EC109 cells, upon knockdown of PTBP1. Middle panel: qPCR analysis showing the knockdown efficiency of PTBP1 in the indicated cells. Right panel: qPCR analysis illustrating the efficiency of ADAR1 overexpression in the indicated cells. **b**, **c**, **e**, **f**, **g** Each dot represents the mean of technical triplicates. Data are presented as mean ± S.D. of 3 biological replicates (paired, two-tailed Student's *t*-test. n.s., not significant; *P < 0.05; **P < 0.01). **h** Number of editing-dependent ARcircs of which flanking introns have editing-mediated changes in the binding sites of each RBP. Black dots indicate RBPs included in this analysis and the number in the bracket denotes the number of circRNAs with altered binding motifs of the corresponding RBP due to editing in flanking introns. Those which have been previously reported to regulate circRNA biogenesis are highlighted in blue. PTBP1 is highlighted in red. Exact P values and source data are provided in Source Data file.

elements[9,54]. Interestingly, co-depletion of ADAR1 and DHX9 leads to synergistic effect on circRNA production. This implies the possibility that DHX9 plays as a regulator of circRNA biogenesis by tunning the editing frequency. Besides DHX9, recent studies on other non-ADAR editing regulators[24,55] indicate an

additional layer of editing-dependent regulation of circRNA biogenesis.

Depending on their binding sites along RNA transcripts, ADARs can protect mRNA from degradation, regulate precursor microRNA processing and alter splicing pattern[56]. Herein, we

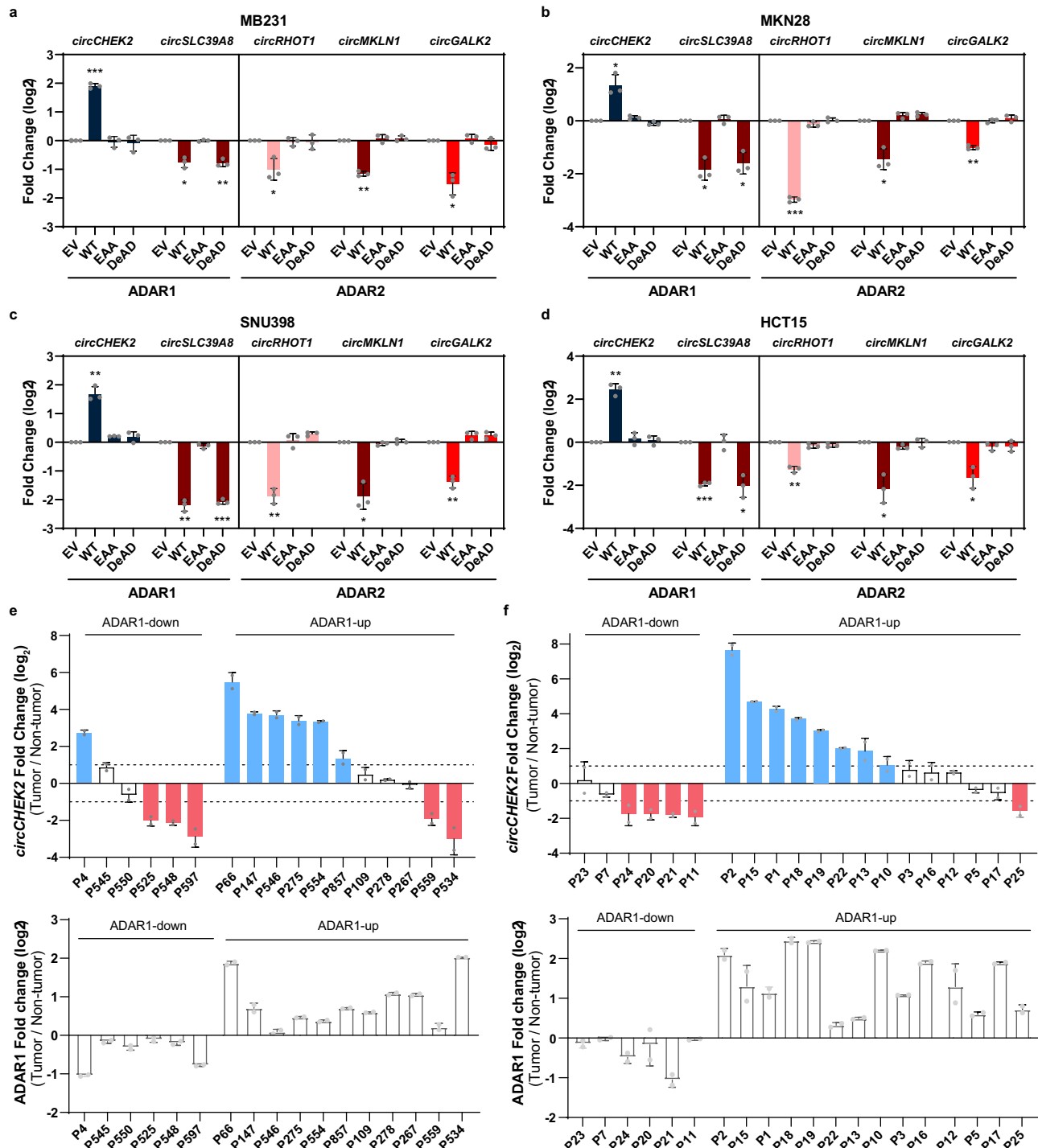

**Fig. 6 ADARs-mediated circRNA regulation exists in multiple cancer types. a–d** Fold change in expression of 5 validated ARcircs upon overexpression of WT, EAA, or DeAD form of ADAR1 or ADAR2, compared to the EV control, in MB231 **a**, MKN28 **b**, SNU398 **c**, and HCT15 **d**. Each dot represents the mean value of technical triplicates. Data are presented as the mean ± S.D. of 3 biological replicates. Statistical significance was calculated by paired, two-tailed Student's *t*-test; *, *P* < 0.05; **, *P* < 0.01; ***, *P* < 0.001. **e**, **f** Fold change in expression levels of *circCHEK2* (upper panels) and *ADAR1* (lower panels) between 17 primary HCC tumors **e** and 20 primary CRC tumors **f** and their matched NT liver and colon samples. Upper panels, cases demonstrating ≥2 fold higher or lower *circCHEK2* expression than their matched NT samples are shown by blue or red bars, respectively. Lower panels, patients were stratified into 2 groups: 'ADAR1-up' and 'ADAR1-down', according to the pattern of change in *ADAR1* expression between HCC or CRC and their matched NT samples. Each dot represents the mean value of technical triplicates. Data are presented as the mean ± S.D. of 2 independent experiments. Exact P values and source data are provided in Source Data file.

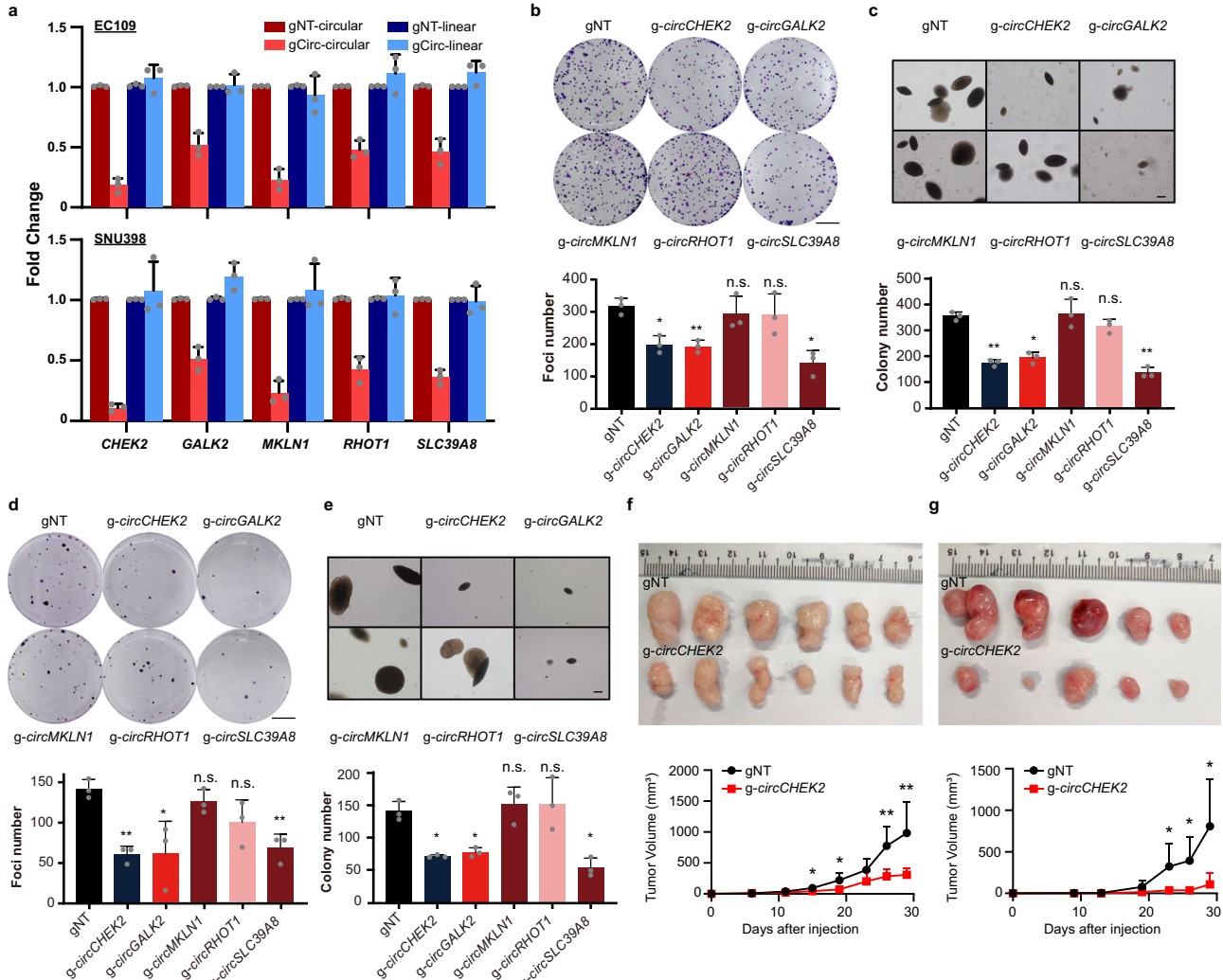

**Fig. 7 ADARs-regulated circRNAs affect tumorigenesis. a** Fold change in expression of circRNAs (circular) and their corresponding host genes (linear) in EC109 (upper panel) and SNU398 cells (lower panel), upon CasRX and guide RNA-mediated knockdown of circRNAs (gCirc) compared to non-targeting control (gNT). gNT-circular, circRNA expression of indicated gene upon treatment of CasRX and non-targeting guide RNA, and so forth. **b, c** Quantification of foci (**b**; scale bar, 1 cm) and colonies formed in soft agar (**c**; scale bar, 200 μm) by EC109 cells with or without circRNAs knockdown. **d, e** Quantification of foci (**d**; scale bar, 1 cm) and colonies formed in soft agar (**e**; scale bar, 200 μm) by SNU398 cells with or without circRNAs knockdown. **f, g** Images and growth curves of xenograft tumors derived from EC109 (n = 6) (**f**) and SNU398 cells (n = 5) **g**. Data are presented as the mean ± S.D. of tumor volumes. Statistic significance was calculated by unpaired, two-tailed Student's *t*-test; *, P < 0.05; **, P < 0.01. **a–e** Each dot represents the mean value of technical triplicates. Data are presented as the mean ± S.D. of 3 biological replicates. Statistical significance was calculated by paired, two-tailed Student's *t*-test; n.s., not significant; *, P < 0.05; **, P < 0.01; ***, P < 0.001. Exact P values and source data are provided in Source Data file.

mapped the identified RNA editing sites by RNA-Seq as well as RCMs of ARcircs to the region near the back-splicing sites. Our results implied the importance of base-pairing of flanking introns in circRNA biogenesis, consistent with previous studies[7,8]. However, we found that unlike RNA binding, RNA editing is not always required for ADARs-mediated regulation of circRNAs. Nevertheless, the precise editing-independent mechanisms remain further investigations.

Circular RNAs were long considered as by-products of aberrant splicing without biological functions. Only until recently, plenty of circRNAs were found to play critical roles in multiple aspects of cellular and physiological functions, and the dysregulated circRNAs have been implicated in tumorgenesis[3,57]. However, how ADARs-mediated changes in circRNA production contribute to cancer remain elusive. Herein, we provided extensive evidence supporting that the editing-dependent and/or independent regulation of circRNA expression by ADAR proteins is present in multiple types of

cancer cell lines and more importantly, the association between expression levels of ADARs and *circCHEK2* could also be found in HCC and CRC patient samples. We further showed these ARcircs are not merely by-products but indeed affect tumorigenesis, which poses an additional important function of ADARs in cancer. Of note, in our study, *circSLC39A8* exhibits an oncogenic role; however, the biogenesis of *circSLC39A8* could be repressed upon overexpression of ADAR1 which is largely characterized as an oncogene[56]. In fact, the tumor promoting effect of ADAR1 is most likely arising from functional changes of multiple target genes via ADAR1's editing-dependent and independent functions. The effect of ADAR1 or ADAR2 on tumorigenesis may differ depending on their target genes and/or cell or tissue types. For example, ADAR1-mediated protein-recoding editing of antizyme inhibitor 1 (*AZIN1*) promotes hepatocarcinogenesis[21]; however, ADAR1-mediated editing can also suppress tumorigenesis by recoding Gamma-aminobutyric acid receptor subunit alpha-3 (*GABRA3*) in breast cancer[58]. Likewise,

each ARcirc may have its distinct cancer-related functions and thus, one should note that contribution of ADARs-regulated circRNA biogenesis to cancer is unlikely to be attributed to one single ARcirc.

In sum, by identifying more than a thousand circRNAs regulated by ADARs, we uncover that ADARs could regulate circRNAs in both direction via editing-dependent and editing-independent mechanisms. We provide experimental evidence that ADAR1/2 can edit RCM of ARcirc, altering the secondary structure formed between RCMs within the flanking introns and enhanced binding of RBP to the site of action. Moreover, ADARs-mediated circRNA regulation is most likely to be present in the same manner across different types of cancer cells, including breast, esophagus, liver, stomach and colon. We show that these ARcircs were not merely by-products of back-splicing, but functional molecules influencing tumorigenesis. These findings improve our understanding of the interaction between ADARs and circRNA biogenesis and its biological importance, particularly in the context of cancer.

## Methods

**Ethical statement.** Our research complies with all relevant ethical regulations. All animal experiments were approved by the Institutional Animal Care and Use Committees of National University of Singapore (NUS; Singapore) with the protocol numbers R16-1644 and R20-1586. All human tissue samples used in this study were approved by the committees for ethics review at Sun Yat-Sen University, the National University of Singapore, and the National University Hospital, Singapore. Written informed consent for all patients were provided for the use of their clinical specimens for medical research.

**Cell lines.** EC109 cells were kindly provided by Professor TSAO, George Sai Wah (Director, Faculty Core Facility, Li Ka Shing Faculty of Medicine); SNU-398, HCT15 and MB231 cells were purchased from the American Type Culture Collection (ATCC); MKN28 cells were obtained from Japanese Collection of Research Bioresources Cell Bank. MB231 cells were maintained in Dulbecco's Modified Eagle Medium (Biowest) while the rest cell lines were maintained in Roswell Park Memorial Institute (RPMI) medium (Biowest), all supplemented with 10% fetal bovine serum (FBS; Thermo Scientific) and incubated at 37 °C with 5% $CO_2$.

**Identification of circRNAs by circRNA sequencing.** The expression level of ADAR1 or ADAR2 was modulated (forced overexpression or silencing) using a lenti-viral system in EC109 cells. RNA extraction was performed using RNeasy Mini kit (Qiagen) and subjected to rRNA depletion and RNase R treatment to digest all the linear RNAs. Samples were purified with Beckman RNAClean beads, the retrieved RNA was fragmented using divalent cations at an elevated temperature. Libraries were prepared using the TruSeq library preparation protocol (Illumina) using a modified protocol. Fragments were purified with Beckman AMPure beads and resolved in EB buffer for end repair and adding A at the 3' end. Y-adaptor was added afterwards. The product was then amplified to construct the cDNA library and sequenced on an Illumina HiSeq4000 or NovaSeq6000 instrument.

High-confidence circRNAs were identified by using an established in-house bioinformatics pipeline[59]. First, raw reads were mapped to the reference human genome (*hg19*) by STAR (v2.5.2a)[60] with the chimeric junction reads option on (*--chimSegmentMin 20*). The gtf file from the GENCODE[61] (*gencode.v27lift37.annotation.gtf*) was used for the gene and junction annotations. The expression of the circRNAs were quantified as read counts that map to the junctions, resulting in a total of 37,916 circRNAs having > = 1 read in at least 1 out of the 3 samples. To identify differentially expressed circRNAs in ADAR1/2 overexpressed and silenced samples compared to the control, the below criteria were applied:

1. Total reads in both (EV + ADAR OE) and (Scr + ADAR KD) > = 10,
2. Fold change in circRNA expression (ADAR versus EV) < = 0.5 (suppressing) or > = 2 (promoting), and
3. Fold change in circRNA expression (shADAR versus shScr) <= 0.5 (promoting) or > = 2 (suppressing).

As a result, we identified 1,406 circRNAs which are potentially regulated by ADARs (Supplementary Data 1).

To conduct a reliable comparison of the performance for circular RNA identification between our in-house pipeline and the other two commonly used benchmark methods- CIRI2 (v2.0.6) and CIRCexplorer2 (v2.3.0)[26–28], we considered annotated circRNAs wherever possible and required the junction positions to be identical (chr-start-end).

To identify editing-dependent ARcircs using circRNA-Seq datasets of the DeAD mutant or EV control-overexpressing EC109 cells (DeAD or EV), the following criteria were applied:

(1) Total reads in (EV + DeAD) ≥ 10;
(2) The resultant list of circRNAs was overlapped with ARcircs presented in Fig. 1b and Supplementary Data 1;
(3) We obtained a total of 1,073 ARcircs fulfilling (1) and (2). We next defined "editing-dependent ARcircs" using the following filter criteria: there is no or minor change (0.8 < fold change < 1.25) in expression between the EV and DeAD-overexpressing cells or the pattern of change in expression upon overexpression of DeAD mutants is opposite to that of the corresponding wildtype ADAR1/2 when compared to the EV control. The remaining ARcircs which are 1) regulated by the wildtype or DeAD form of ADAR1 or/and ADAR2 in the same direction and 2) demonstrate ≥1.25-fold change in expression upon overexpression of DeAD mutant versus EV control are defined as "editing-independent ARcircs" (Supplementary Data 3).

**Identification of high-confidence A-to-I editing events from the total RNA sequencing data.** A bioinformatics pipeline adapted from a previously published method[62] was used to identify RNA editing events from total RNA-Seq data by using CSI NGS Portal (https://csibioinfo.nus.edu.sg/csingsportal)[59]. For each sample, raw reads were mapped to the reference human genome (*hg19*) with a splicing junction database generated from transcript annotations derived from UCSC, RefSeq, Ensembl and GENCODE[61] by using Burrows–Wheeler Aligner with default parameters (*bwa mem*, v0.7.17-r1188)[63]. To retain high quality data, PCR duplicates were removed (*samtools markdup -r*, v1.9)[64] and the reads with mapping quality score < 20 were discarded. Junction-mapped reads were then converted back to the genomic-based coordinates. An in-house perl script was utilized to call the variants from samtools pileup data and the sites with at least two supporting reads were retained. The candidate events were filtered by removing the single nucleotide polymorphisms (SNPs) reported in different cohorts (1000 Genomes Project[65], NHLBI GO Exome Sequencing Project, and dbSNP v138[66]) and excluding the sites within the first six bases of the reads caused by imperfect priming of random hexamer during cDNA synthesis. For the sites not located in Alu elements, the candidates within the four bases of a splice junction on the intronic side, and those residing in the homopolymeric regions and in the simple repeats were all removed. Candidate variants located in the reads that map to the non-unique regions of the genome by using BLAST-like alignment tool[67] were also excluded. At last, only A-to-G editing sites based on the strand information from the strand-specific RNA-Seq data were considered for all the downstream analyses. The genomic regions of the editing variants and the associated genes were annotated by using ANNOVAR (v2018)[68] with the *refGene* table. As reported by us previously[15], ADARs-mediated global editing changes has been confirmed by analysing the same RNA-Seq dataset of ADARs OE or KD cells.

To identify high-confidence editing events, the editing sites were required to be supported by ≥10 reads in ≥1 sample, and ADAR1/2-overexpressing samples to result in more than 10% change in the editing level compared to the control. This resulted in 41,151 high-confidence editing sites from the RNA-Seq of our EC109 cell lines used in further analyses. To analyze the sequence preference for the neighbour nucleotide surrounding editing sites, the sequence context of these editing sites was extracted using "bedtools getfasta", i.e. editing site plus 2 neighbour nucleotides on either side in a strand-specific manner. Then the nucleotide frequencies were converted to a position probability matrix and the sequence logo was plotted by using the "seqLogo" package (v1.56.0).

**In silico prediction of RCMs within flanking intronic sequences.** To identify RCMs, we adopted a previously published method[8]. We used circRNAs exhibiting >2 or <0.5-fold change upon ADAR1 or ADAR2 overexpression with total reads in EV and ADAR1/2 more than 50 in circRNA-seq for RCM identification, which results in 1,118 circRNAs in total (Supplementary Data 2). First, a BLAST[69] alignment was performed for each intron pair flanking circRNA junctions to identify all the potential candidates. The RCM with the top BLAST score for each circRNA was retained yielding 1,043 circRNAs with at least 1 RCM. The circRNAs with short flanking introns (< 1500 bp) were further removed leaving 886 circRNAs as the final list. The intronic region spanning 1500 bp upstream and 1500 bp downstream of the circRNA back-splicing junctions was considered to plot the distribution of RCM coverage, which is defined as the sum of the number of top-scoring RCMs at each base across all 886 circRNAs.

To investigate if there is an enrichment of RNA editing events within RCMs as previously reported[8], 41,151 high-confidence RNA editing sites identified from the RNA-Seq data were overlapped with the flanking introns of 886 circRNAs identified from the circRNA-Seq. The density plots showing the distribution of editing sites were generated within the same region as the plot illustrating the distribution of the RCM coverage mentioned above, but we used the density distribution rather than coverage since each editing site indicates a single nucleotide variation whereas RCMs are regions of variable length. In addition, we have also analyzed the distribution of 2,576,459 A-to-I RNA editing sites downloaded from RADAR database[36] within the same regions.

**Plasmid constructions.** Minigene fragments were amplified from human placenta genomic DNA (Sigma) (for intronic sequences) or EC109 cDNA (for exonic sequences) using PrimeSTAR Max DNA polymerase (Clontech) with overlapped

primers and ligated into one piece of DNA, followed by ligation into pcDNA3.1+ vector. KAPA HiFi polymerase (KAPA Biosystems) was used to introduce point mutations into minigene using primers with corresponding mutation(s).

Overexpression plasmids were obtained by cloning coding sequences of protein, which were amplified by PrimeSTAR Max DNA polymerase (Clontech), into pLenti6 vector. ADARs-targeting short hairpin RNAs (shRNAs) were designed using RNAi Platform (Broad Institute) and were cloned into pLKO.1_puro plasmid using AgeI and EcoRI restriction sites.

CasRX system (pXR001: EF1a-CasRx-2A-EGFP and pXR003: CasRx gRNA cloning backbone) was a gift from Patrick Hsu (pXR001: Addgene plasmid #109049, http://n2t.net/addgene:109049, RRID:Addgene_109049; pXR003: Addgene plasmid #109053, http://n2t.net/addgene:109053, RRID:Addgene_109053)[51,52]. Guide RNA (gRNA) sequences were designed using sequence of circRNAs around BSJ with a length of 21 bp and cloned into pXR003 plasmid using BbsI restriction sites.

**Plasmids transfection**. A total of 2 μg plasmids (protein overexpression construct, shRNA plasmids or minigene plasmids) or a mixture of 1ug pXR001 and 1ug pXR003 plasmids were transfected into cells a well of 12-well plate using Lipo-fectamine 2000 (Invitrogen) with a ratio of 1:2 (DNA:reagent).

**RNA extraction and RT-qPCR**. RNA was extracted using RNeasy mini kit (Qiagen) with on column treatment of DNaseI. cDNA was synthesized using Advantage RT-for-PCR kit (Clontech) with random hexamer primers and subsequently qPCR was performed with GoTaq DNA polymerase (Promega). Fold change was calculated by $2^{\wedge}\text{-}\Delta\Delta Ct_{sample}$. $\Delta Ct = Ct_{target} - Ct_{actin}$; $\Delta\Delta Ct = \Delta Ct_{sample}$-average$\Delta Ct_{control}$. Primers used in RT-qPCR are listed in Supplementary Data 5.

**Western blot**. Cells were lysed with RIPA buffer (Sigma) supplemented with 1x cOmplete EDTA-free protease inhibitor cocktail (Roche) and concentrations of total protein were quantified using Bradford assay (Biorad). 10% SDS-PAGE were used to separate proteins, followed by transferred onto polyvinylidene difluoride membranes (Millipore) and incubated with primary antibodies (1:1,000 dilution) overnight at 4 °C and secondary antibodies (1:10,000 dilution) at room temperature for 1 h. Enhanced chemiluminescence (GE Healthcare) was used to visualize the blots. Primary antibodies used in this study are as listed: anti-PTBP1 (Abcam, ab133734), anti-TDP43 (Proteintech, 10782-2-AP), anti-ADAR1 (Abcam, ab88574), anti-FLAG-HRP (Sigma, A8592), anti-β actin HRP (Santa Cruz Bio-technology, sc-47778HRP), anti-mouse IgG HRP-linked (Cell Signaling Technol-ogy Cat# 7076, RRID:AB_330924), anti-rabbit IgG HRP-linked (Cell Signaling Technology Cat#7074, RRID:AB_2099233). ImageJ (1.51J8) was used to measure band density of blots.

**RNA immunoprecipitation (RIP)**. A 10-cm dish of EC109 cells was transfected with 10 μg of FLAG, FLAG-ADAR1, plenti6, or plenti6-ADAR1 plasmid indivi-dually. After 48 h culturing, cells were collected and lysed in buffer containing 50 mM Tris, pH7.5, 150 mM NaCl, 1 mM EDTA and 1% Triton X-100 supple-mented with cOmplete protease inhibitor (Roche) and SUPERase·In RNase Inhi-bitor (Invitrogen). For FLAG-RIP, lysates were then incubated with anti-FLAG M2 magnetic beads (Sigma) overnight at 4 °C with rotation followed by six times of washing with 1× TBS buffer (0.5 M Tris, 1.5 M NaCl). For PTBP1 RIP, the lysates were pre-cleared using 50 μL protein A-agarose suspension (Roche) at 4 °C for overnight. A total of 2 μL anti-PTBP1 antibody was then added into the lysate and incubate at 4 °C for 1 h, followed by overnight incubation at 4 °C with the addition of 50 μL protein A-agarose suspension. The beads were then washed with washing buffer [150 mM NaCl, 0.04 U/μL SUPERase·In RNase Inhibitor (Invitrogen)] for 6 times with each time for 10 min at 4 °C. 10% of beads was used for protein elution while the rest was subjected to RNA extraction using RNeasy miniprep kit (Qia-gen). Extracted RNA was reverse-transcribed using Advantage RT-for PCR kit (Clontech) with random hexamer and subsequently qPCR was performed. Input indicates 1% of the total cell lysate. %input = $2^{-\Delta Ct} \times 100\%$; $\Delta Ct = Ct_{RIP} - [Ct_{input} -$ dilution factor]. Sequences of primers are listed in Supplementary Data 5.

**Analysis of editing frequency by TA cloning**. The region containing editing site(s) was amplified using PCR method, followed by purification using PCR product purification kit (Qiagen). Purified PCR products were then ligated into pGEM-T easy vector (Promega) using T4 quick ligase (Promega). A total of 20–28 individual plasmids were sent for Sanger sequencing. The number of unedited 'A' or edited 'G' clones was counted, followed by the calculation of the percentage of edited clones by 'G/(A + G)'. The percentage of edited clones (a readout of 'editing frequency') was determined and shown by pie chart (represented by red slice).

**In vitro transcription**. PCR was used to generate DNA template for in vitro transcription with a primer pair containing T7 promoter sequences (5′-CGAAATTAATACGACTCACTATAGG at forward) and sequence of interest. DNA template was subjected to in vitro transcription with RiboMAX™ Large Scale RNA Production Systems (Promega) according to the manufacturer's protocol. Synthesized RNA probes were then purified by RNeasy mini kit (Qiagen).

**Native PAGE analysis**. Each RNA probe (50 pmol) was first dephosphorylated using rSAP (New England Biolabs), followed by $^{32}$P labelling with γ-$^{32}$P-ATP (Perkin Elmer) and T4 PNK (New England Biolabs). Labelled probes were then purified by G25 column (GE healthcare). 0.5 pmol labelled RNA probe were incubated at 95℃ for 5 min and then gradually cool down to help form secondary structure. Probes were then loaded on 4% or 8% native polyacrylamide gel, fol-lowed by gel drying and gel exposure to BioMax® MS film (Carestream Kodak). Sequences of probes are listed in Supplementary Data 5. For *circCHEK2*, position of probe with lowest migration rate was labelled as 0, while position of probe with the highest migration rate was labelled as 1. The related migration rate of each probe was measured by: (Distance between the probe with 0)/(Distance between 1 and 0).

**Whole cell extraction**. Whole cell extraction was performed with the kit (Active motif) according to the manufacturer's protocol.

**RNA pull-down assay**. RNA probe was generated as discussed above but only with an addition of 3′-aptamer at reverse primer. For each reaction, 50 μl Dynabeads MyOne C1 (Invitrogen) was used to incubate with 25 μg RNA probe in 300 μl binding buffer (100 mM NaCl, 10 mM MgCl$_2$, 50 mM Hepes, 0.5% Igegal CA-630, and pH 7.4) for 30 min at 4 °C with rotation for probe binding to the beads. Beads were then washed washing buffer (250 mM NaCl, 10 mM MgCl2, 50 mM Hepes, 0.5% Igegal CA-630, and pH 7.4,) for 10 min at 4 °C for three times. 1 mg whole cell extract was supplemented with 4ul 10 mg/ml yeast tRNA (Invitrogen) and SUPERase In (Invitrogen). The mixture was added to the beads and topped up to 300 μl with washing buffer, followed by incubation for 30 min at 4 °C with rotation. After washing three times, beads were subjected to 2× Laemmli buffer (Sigma) at 95 °C to elute proteins followed by western blot (WB). Sequences of probes are listed in Supplementary Data 5.

**RBPmap analysis**. The ±10nt sequences surrounding editing sites were retrieved. The sequence with A or G at each editing site was used as input for RBP motif analysis using RBPmap[46]. Briefly, for each sequence, if the RBP binding affinity (Z-score) is changed because of an A-to-G mutation, the RBP binding motif is affected by the editing site. The number of circRNAs which have altered RBP binding sites on flanking introns due to editing was calculated using our in-house script.

**Foci formation assay**. A total of 1,000 cells were seeded in each well of 6-well plates after transfection and incubated at 37 °C for 7-9 days. Plates were stained with crystal violet solution (0.1% crystal violet, 20% methanol in PBS) to visualize colonies. Colonies were calculated using OpenCFU[70]. Triplicate independent experiments were conducted with technical triplicates.

**Soft agar assay**. A total of 2,000 cells (EC109) or 5,000 cells (SNU398) were seeded into 0.4% low-melting agarose (Lonza Rockland) in each well of 6-well plates with 0.6% low-melting agarose at the bottom. Plates were incubated at 37 °C for 2 weeks and stained with crystal violet solution (0.05% crystal violet, 20% methanol in PBS) for visualization. Colonies were calculated using OpenCFU[70]. Triplicate independent experiments were conducted with technical triplicates.

**In vivo tumorigenicity assay**. Four-six-weeks-old male and female NOD scid gamma (NSG) mice (The Jackson Laboratory, RRID:IMSR_JAX:005557) were maintained in pathogen–free (SPF) facility in NUS Comparative Medicine Department. Less than 5 mice with same sex were housed in a cage at 20–25 °C and 50% humidity with a 12 h light/dark cycle. A total of 6 female mice (EC109) or 4 female + 1 male mice (SNU398) were used to subcutaneously inject with one million (EC109) or two million (SNU398) cells into the right and left flanks. Tumor growth was monitored by measuring the length and width at indicated day points. Tumor volume was determined by the formula: $0.5 \times \text{length} \times \text{width}^2$. All animal experiments were approved by and performed in accordance with the Institutional Animal Care and Use Committees of National University of Singapore (NUS; Singapore).

**Human tissues**. A total of 17 matched pairs of primary HCC and adjacent non-tumor (NT) tissues were obtained from the Sun Yat-Sen University Cancer Centre (Guangzhou, China), between 2002 and 2007. A total of 20 matched pairs of primary CRC and adjacent NT colon tissues were obtained from the National University Hospital, Singapore.

**Quantification and statistical analysis**. All quantitative data represent the mean ± SD. Statistical significance was accessed with paired or unpaired two-tailed Student's *t*-tests using Prism 8 (GraphPad software). For all figures: n.s., not sig-nificant; *, $P < 0.05$; **, $P < 0.01$; ***, $P < 0.001$.

**Reporting summary**. Further information on research design is available in the Nature Research Reporting Summary linked to this article.

## Data availability

The circRNA-Seq data generated in this study have been deposited in the Gene Expression Omnibus (GEO) under accession code GSE164681. The EC109 RNA-Seq data have been published previously and is also available at GEO under accession GSE131658[15]. Human genome reference hg19 was obtained from GENCODE. A-to-I editing sites from RADAR database were obtained from http://RNAedit.com. Information on SNPs was obtained from 1000 Genomes Project (https://www.internationalgenome.org/), NHLBI GO Exome Sequencing Project, and dbSNP v138. The data supporting the findings of this study are available from the corresponding authors upon reasonable request. Source data for the figures and supplementary figures are provided as a Source Data file. Source data are provided with this paper.

## Code availability

The codes used in the data analysis are available in Supplementary Software.

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

## Acknowledgements

We thank and acknowledge Prof. Xin-Yuan Guan (The University of Hong Kong, Hong Kong, China) for providing RNA samples of HCC cases. L. C. was supported by National Research Foundation Singapore; Singapore Ministry of Education under its Research Centres of Excellence initiative; Singapore Ministry of Education's Tier 2 Grants [MOE2018-T2-1-005 and MOE2019-T2-2-008]; NMRC Clinician Scientist-Individual Research Grant (CS-IRG, project ID: MOH-000214); and Singapore Ministry of Education's Tier 3 Grants [MOE2014-T3-1-006].

## Author contributions

L.C. conceived and supervised the study. L.C. and H.S. designed and performed the experiments. H.Y., O.A. and X.R. conducted all the bioinformatics analyses. Y.S. helped in conducting primary tumor sample related experiments. K.T. provided the matched pairs of CRC and NT samples. Y.S. and X.K. assisted in conducting mouse-related experiments. X. K. helped with data analysis. S.J.T., J.H., X.K., D.J.T.T., V.H.E.N., F.B.M., P.P., and K.W.L. provided insightful suggestions and experimental materials. H.S. and O.A. wrote the manuscript. L.C. edited the manuscript.

## Competing interests

The authors declare no competing interests.
