## [Peer Review File · Nature Communications]

Title: ADARs act as potent regulators of circular transcriptome in cancerREVIEWER COMMENTS

Reviewer #1 (Remarks to the Author):

The authors examine the effect of ADARs on circRNA levels in cancer cells. They find hundreds of circRNAs whose levels change upon ADAR1 or 2 over-expression and characterize one of these in detail. The manuscript is easy to read but the data presentation needs to be improved and some of the conclusions are premature.

Main concerns:

- (1) Rigor of data shown needs to be improved. In almost all of the graphs, the authors are showing the results of “technical triplicates from a representative experiment of 2 independent experiments.” This means the error bars only indicate the accuracy of pipetting on a given day, and not the critical point data reproducibility from day to day. The authors need to always show data from 3 biological replicates throughout the manuscript (Fig 1e,f; Fig 2c, d; Fig 3d,e; Fig 5c-e; Fig 6c-f).
- (2) Details of the circRNA annotation are limited, but it appears a single algorithm has been used. It is well established that circRNA annotation methods have high false positive rates (see work from Thomas Hansen) so an additional annotation pipeline needs to be used to ensure that all circRNAs are “high confidence.”
- (3) After Fig 1a, the authors tend to look at cells in which non physiologically relevant levels of ADAR1/2 are present (200-500 fold over-expression in Supp Fig 2). This raises some questions of how important editing is in wild type cells with endogenous levels of ADAR1/2. I was surprised the authors have not done more work with knockdown of ADAR1/2.
- (4) Fig 3c: The authors have made a circCHEK2 minigene but have not adequately examined its outputs. It is possible that this construct makes linear concatemers or is subjected to trans-splicing which would yield qPCR products indistinguishable from circRNAs. Non-PCR based approaches, e.g. northern blotting, should be used to examine the outputs of this minigene plasmid.
- (5) The validation experiments were performed on a single gene (circCHEK2). The authors have made predictions for how editing of specific bases could affect several other circRNAs (Fig 4c,d) but none of this has been tested. It remains unclear if the predictions are or are not accurate.
- (6) Building off of point 5, it would greatly strengthen the manuscript if the authors could make large-scale predictions of effects on circRNA levels based on the exact positions of editing sites and the localized secondary structure. The authors have provide nice ideas about how editing can sometimes strengthen hairpins and hence promote backsplicing, but it is unclear if these ideas are true beyond the circCHEK2 gene.

(7) Fig 5b needs to be quantified with error bars.

(8) Fig 5c: The authors need to confirm that the expression of the linear RNA is not changed in these experiments.

(9) P. 9: "Taken together, editing can not only alter the stability of secondary structure formed between RCMs, but also affect RBP binding to flanking intronic sequences, leading to changes in circRNA production." This claim would be more justified if the authors could use CLIP and show that RBP binding to the intron is changed in cells, not just in vitro as was done in Fig 5b

Minor points:

(1) Page 3: "covalently closed continent loop structure" should be "covalently closed continuous loop structure"

(2) Page 3: "The first circRNA, which is a viroid in plants, was discovered and described in 1976." In the abstract, the authors have defined circRNAs as being produced by back-splicing and viroids are not made in this manner.

(3) Page 4: The high expression of ADARs and circRNAs in the brain is a weak argument for why to study the connection between these molecules.

(4) Page 10: Clarify in main text that a Cas13 method is being used.

Reviewer #2 (Remarks to the Author):

The authors provided experimental evidence to support the hypothesis that A-to-I RNA editing can affect circRNA formation in a bidirectional manner, in which RNA editing alters the secondary structure formed between reverse complementary sequences residing in the introns flanking circularized exons through correcting A:C mismatches to I(G)-C pairs or creating I(G).U wobble pairs. They also found that RNA editing can affect RBP (e.g., PTBP1) recruitment and thereby regulate circRNA biogenesis. Finally, they showed that some ARcircs indeed influence tumorigenesis in multiple cell types. Their results modified the previous opinion that ADARs function as suppressor of circRNA formation by editing and melting the dsRNA. Some comments are listed below.

1. For identification of "high-confidence" A-to-I RNA editing events, the authors should evaluate the specificity of the identified sites by measuring (1) the fraction of A-to-G mismatches to all types of mismatches (%AG), (2) the ratio of #G-to-A mismatches to #A-to-G mismatches, and (3) the cis-preference of ADARs (the presence of "G" at the 5' and 3' neighbor nucleotides next to the A-to-I editing sites).

2. The following analyses should be included in the study: (1) the changes of editing levels after ADAR OE or ADAR KD; (2) the number of high-confidence A-to-I editing sites correcting A:C mismatches or creating wobble pairs within the RCMs regions; and (3) the expression levels of circRNAs affected by

such an editing-dependent mechanism.

3. The detailed information of the input sequence for RNA structure prediction should be provided. Did the input sequence include circularized exons?

4. For validations of the selected back-splicing events, all the Sanger sequencing results of the 21 cases should be provided (only 6 cases were illustrated in Supplementary Fig. 1c).

5. Whether ADAR1 “indeed” binds and edits the dsRNA structure formed between RCMs located in the flanking introns of circCHEK2 should be experimentally confirmed.

6. In Fig. 3b (pie charts), how did the authors get the editing levels of the three sites from endo RCM and minigene RCM? Were the editing levels validated by Sanger sequencing?

7. In figure 5b, the probe without RBP binding sites should be added as a negative control. In addition, to confirm the physical interactions between RBPs and the intron RCMs, the RNA immunoprecipitation experiments based on anti-RBPs should be performed. The binding affinity could be measured after ADAR OE or ADAR KD.

8. PTBP1 may affect ADAR1 translation (*Cellular and Molecular Life Sciences* 72.22 (2015): 4383-4397). Thus, it is important to explore the changes of ADAR1 protein expression after PTBP1 knockdown to rule out this possibility.

9. In Fig. 6, it would be better to discuss why ADAR1 down-regulated circSLC39A8 exhibited the reduced tumorigenic ability upon circSLC39A8 knockdown. The difference of ADAR expression or editing capability between cancer and normal cells should be analyzed.

10. Page 10, typo: “SNU198”  SNU398?

Reviewer #3 (Remarks to the Author):

In this study, Shen et al. characterized ADARs as potent regulators of circular transcriptome through stabilizing or destabilizing secondary structures and/or changing the RBP binding. The authors further showed the functional effects of ARcircs in multiple cancers and highlighted the complexity of cross-talk in RNA processing and its contributions to tumorigenesis. The manuscript is overall interesting, significant, and well-organized. The reviewer has a few comments/suggestions:

1. It is interesting to show that A-to-I editing alters circRNA production via stabilizing or destabilizing dsRNA formed between RCMs (Fig. 4a). Could the authors apply similarly *in silico* secondary structure for all ARcircs to examine that if this can explain the distribution of promoted and suppressed circRNAs presented in fig. 1b?

2. It is significant to conclude that ADARs-mediated circRNA regulation exists in multiple cancer types. The authors showed very nice experimental data in multiple cancer cell lines. Is it possible for the authors to confirm this pattern in patient data (author's own data or public data) – for example, compare the ARcircs expression in ADAR high vs. ADAR low expression samples.

3. It is interesting to show that two RBPs with a respective binding motif near circCHEK2. How about other RBPs for other ARcircs? Since recent studies showed that other RBPs may also regulate the circRNA expression (e.g., PMID: 31446897). A computational analysis regarding other RBPs/circRNAs will enhance the conclusion.

4. The authors mentioned the controversial results from previous studies (references 19, 20, 24) in the discussion. Could the authors explain the different observations?

5. GSE164681 is not accessible.

Reviewer #1:

The authors examine the effect of ADARs on circRNA levels in cancer cells. They find hundreds of circRNAs whose levels change upon ADAR1 or 2 over-expression and characterize one of these in detail. The manuscript is easy to read but the data presentation needs to be improved and some of the conclusions are premature.

Main concerns:

(1) Rigor of data shown needs to be improved. In almost all of the graphs, the authors are showing the results of “technical triplicates from a representative experiment of 2 independent experiments.” This means the error bars only indicate the accuracy of pipetting on a given day, and not the critical point data reproducibility from day to day. The authors need to always show data from 3 biological replicates throughout the manuscript (Fig 1e,f; Fig 2c, d; Fig 3d,e; Fig 5c-e; Fig 6c-f).

-Thank the reviewer for pointing it out. We have addressed the reviewer’s concern. Please see the details below:

- In the revised **Fig. 2c, d; Fig. 3b (new data), e, f; Fig. 5b-c (new data) and e-g; Fig. 6a-c, d (new data); Fig. 7a-e; Supplementary Fig. 1c, f-g (new data); Supplementary Fig. 2c, d; Supplementary Fig. 3b-d (new data); Supplementary Fig. 4a, b (new data)**, each dot represents the mean value of technical triplicates from an independent experiment and data are shown as the mean \pm SD of 3 biological replicates (i.e., 3 independent experiments).
- The data shown in the original **Fig.1e-g** and **Supplementary Fig. 1e** were obtained from the leftover RNA samples after the majority was sent out for the total and circular RNA-Seq analyses. We only intended to use these leftover original samples (without including biological replicates) to experimentally validate the differential expression of candidate circRNAs detected by circular RNA-Seq. In fact, the biological replicates have been included in the following experiments shown in **Fig. 2c, d**.

(2) Details of the circRNA annotation are limited, but it appears a single algorithm has been used. It is well established that circRNA annotation methods have high false positive rates (see work from Thomas Hansen) so an additional annotation pipeline needs to be used to ensure that all circRNAs are “high confidence.”

-According to the reviewer’s suggestion, we have selected CIRI2 and CIRCexplorer2 as 2 additional tools/pipelines due to the following 2 reasons:

1. CIRI2 and CIRCexplorer2 are among top 5 mostly used benchmark methods for circular RNA identification (Chen et al., 2021; Gao et al., 2018; Zhang et al., 2016).
2. CIRI2 and CIRCexplorer2 use different aligners, which ensures a fair and reliable comparison. CIRCexplorer2 uses the same aligner (STAR) as our pipeline, while CIRI2 utilizes different aligner (BWA).

In addition, to ensure the accuracy of our comparison, we have considered annotated circRNAs wherever possible and required the junction positions to be identical (chr-start-end). Overall, we have obtained a high percentage of overlapping circRNAs between our in-house pipeline and either of these 2 tools (**87%** for CIRCexplorer2; **70%** for CIRI2). We have added this data into the new **Supplementary Fig. 1a** and provided the relevant information in the “Results” and “Methods”.

tool	aligner	total (n)	overlap (n)	overlap (%)
our pipeline	STAR	37916	37916	100.0%
CIRCexplorer2	STAR	40956	35693	87.2%
CIRI2	BWA	25446	17788	70.0%

(3) After Fig 1a, the authors tend to look at cells in which non physiologically relevant levels of ADAR1/2 are present (200-500 fold over-expression in Supp Fig 2). This raises some questions of how important editing is in wild type cells with endogenous levels of ADAR1/2. I was surprised the authors have not done more work with knockdown of ADAR1/2.

- Thank the reviewer for pointing it out. We have provided new data showing the differential expression of ADAR1/2-regulated circRNAs can be validated in ADAR1/2-depleted EC109 cells (revised **Supplementary Fig. 1f, g**).

(4) Fig 3c: The authors have made a *circCHEK2* minigene but have not adequately examined its outputs. It is possible that this construct makes linear concatemers or is subjected to trans-splicing which would yield qPCR products indistinguishable from *circRNAs*. Non-PCR based approaches, e.g. northern blotting, should be used to examine the outputs of this minigene plasmid.

-We note the reviewer's concern here and agree that by doing northern blotting, we can determine all possible transcripts of different length using a probe; however, we are unable to examine the sequence of each transcript detected by northern blotting. After serious consideration, we came up with an alternative approach. We designed primers covering different parts of the minigene (**Rebuttal Fig. 1a**) and then extracted RNA samples from cells expressing *circCHEK2*. Untreated or RNase R-treated RNA samples were used for RT-PCR analysis, followed by PCR product purification and Sanger sequencing.

Using a convergent primer set (F1+R1), we detected two PCR products (product_a and product_b, **Rebuttal Fig.1b, lane 1**). Sanger sequencing result confirmed that product_a was amplified from a full-length transcript with all exons 2-10 included (transcript_a, **Rebuttal Fig. 1c**) while product_b was amplified from a transcript with exons 3-9 skipped (transcript_b, **Rebuttal Fig. 1c**). Using the other 2 convergent primer sets (F1+R2 and F2+R1), we only detected the partial sequences of transcript_a (product_c, product_d) (**Rebuttal Fig.1b, lanes 2,3**). Importantly, when RNA samples were pre-treated with RNase R, all 4 PCR products (product_a, b, c, d) were completely or largely undetectable (**Rebuttal Fig.1b, lanes 5-7**), suggesting that they are linear transcripts. Sanger sequencing results confirmed that none of them contains backsplicing junction sequence. With a divergent primer set (F2+R2), we amplified the product_e from an RNase R-resistant transcript_c (**Rebuttal Fig. 1c; Rebuttal Fig.1b, lanes 4, 8**) and product_e contains the backsplicing junction sequence (**Rebuttal Fig. 1d**). All these data suggested that the *circCHEK2* minigene produces 2 linear transcripts (transcript_a and transcript_b) and one circular transcript_c, which supports our qPCR results.

Rebuttal Figure 1. Analysis of transcripts derived from the *circCHEK2* minigene. **a.** Schematic diagram illustrating the location of primers on *circCHEK2* minigene. **b.** Agarose gel electrophoresis of RT-PCR product(s) amplified with the indicated primer set, before and after RNase R digestion. **c.** Schematic diagram illustrating the three transcripts derived from *circCHEK2* minigene. **d.** Sequence chromatograms showing the sequences at back-splicing junctions (BSJ) of minigene-*circCHEK2*.

(5) The validation experiments were performed on a single gene (circCHEK2). The authors have made predictions for how editing of specific bases could affect several other circRNAs (Fig 4c,d) but none of this has been tested. It remains unclear if the predictions are or are not accurate.

- Thank the reviewer for the valuable suggestion. To address this, we generated RNA probes containing the wildtype (unedited) or edited (A-to-G mutations introduced at corresponding editing sites) RCM sequences for 6 additional editing-dependent ARcircs (circRNAs regulated by ADAR1/2) and performed the native PAGE gel analysis. As expected, for 3 ADAR-promoted circRNAs (*circASHIL*, *circANKLE2-1* and *circRNF114*), the edited probes migrated faster in the gel than the wildtype probes (revised **Fig. 4c** and **Fig. 4e, left panel**); while for 3 ADAR-repressed circRNAs (*circSYNC*, *circDHX34* and *circRHOT1*), the edited probes migrated slightly slower (revised **Fig. 4d** and **Fig. 4e, right panel**). Altogether, we concluded that there could be a universal editing-dependent mechanism by which ADARs regulate circRNA biogenesis via editing-mediated change in the secondary structure formed by flanking introns.

(6) Building off of point 5, it would greatly strengthen the manuscript if the authors could make large-scale predictions of effects on circRNA levels based on the exact positions of editing sites and the localized secondary structure. The authors have provide nice ideas about how editing can sometimes strengthen hairpins and hence promote backsplicing, but it is unclear if these ideas are true beyond the circCHEK2 gene.

- To address this comment, we overexpressed the catalytic mutants of ADAR1 or ADAR2 (ADAR1-DeAD or ADAR2-DeAD) or the empty vector (EV) control in EC109 cells and performed circRNA-Seq to identify editing-dependent ARcircs. First, in this new batch of circRNA-Seq, 76.3% (1,073/1,406) of ARcircs identified from our previous circRNA-Seq could be detected again. Among these 1,073 ARcircs, we defined editing-dependent ARcircs using the following filter criteria: 1) there is no or minor change ($0.8 < \text{fold change} < 1.25$) in expression between the EV and DeAD-overexpressing cells; or 2) the pattern of change in expression upon overexpression of DeAD mutants is opposite to that of the wildtype ADAR1/2 when compared to the EV control. In total, we have obtained 767 editing-dependent circRNAs regulated by ADAR1 and/or ADAR2 (revised **Fig. 2e**). We previously examined the editing dependency for each of 20 candidate ARcircs and identified 11 editing-dependent ARcircs (**Fig. 2c,d**). Intriguingly, 9 out of 11 editing-dependent ARcircs could be identified by our new circRNA-Seq analysis described above, indicating the reliability of our analysis of editing-dependent ARcircs.

We then went on to identify editing sites within RCMs of these 767 editing-dependent ARcircs. Since intronic regions tend to have low coverage in RNA-Seq datasets, when we applied the stringent criteria as described in **Fig. 2a** (total reads ≥ 10 in at least one sample and absolute editing frequency > 0.1), we obtained less than 100 editing sites located in RCMs of 47 editing-dependent ARcircs, which are not sufficient for any following analysis. Therefore, to obtain more editing sites, we applied a less stringent filter criterion (≥ 1 variant read(s) supporting A>G mismatch in any sample) and identified 387 editing sites within RCMs of 89 editing-dependent ARcircs. We then utilized RNAfold to computationally predict the secondary structure formed within RCMs of these editing-dependent ARcircs. Editing sites detected in RCMs of 71 editing-dependent ARcircs were predicted to reside in structured regions formed between RCMs, which allowed us to analyse the opposite nucleotides to the identified editing sites using a published script (Brummer et al., 2017). It is known that free energy of the structure is lowered, and thus, the stability of the structure is increased (Doshi et al., 2004; Wu et al., 2009). RNAfold, which can compute the minimum free energy (MFE), was used to calculate the change in MFE of the predicted secondary structure before and after editing. Correcting A:C mismatch to I-C pair leads to a more severe change on free energy and structure compared to altering A-U pair to I-U wobble pair (Vendeix et al., 2009; Wright et al., 2018). Our analysis confirmed that RCMs containing 1~2 or ≥ 3 editing sites at A:C mismatches (≥ 3 AC, $n=9$; 1 or 2 AC: $n=27$) tend to have a lower free energy post editing; while RCMs do not contain editing site(s) at A:C mismatches but with the majority at A-U pairs (non-AC, $n=35$) tend to have a higher free energy after editing (**Rebuttal Fig. 2a**). In sum, *in silico* analysis suggested that editing can alter the stability of dsRNA structures.

We went on to study the association between editing-mediated structural change and expression change of editing-dependent ARcircs. Among these ARcircs, we did not observe any significant change in expression between AC-containing or non-AC ARcircs (**Rebuttal Fig. 2b**). In fact, it is not surprising to see such data, for the following reasons:

1) Editing may regulate circRNA biogenesis via changing the RBP binding sites and altering the binding affinity of RBPs to the target regions. In this study, we have shown that PTBP1 can preferentially bind to the edited RCM of *circCHEK2* (revised **Fig. 5**). In addition, as requested by reviewer 3, we conducted a large-scale analysis of RBP binding sites within the flanking introns of editing-dependent ARcircs and showed that most RBPs included in the analysis have editing-mediated changes in their binding sites in flanking introns of more than 10 editing-dependent ARcircs, even with the editing sites identified using our stringent criteria (revised **Fig. 5h**).

2) Due to the technical limitations, a lot of intronic editing sites failed to be detected by RNA-Seq. In our study, despite using a less stringent filter, we only detected 387 editing sites in RCMs and thus identified a low number (71) of editing-dependent ARcircs. This significantly affected our large-scale prediction of effects on circRNA levels based on the exact positions of editing sites and the localized secondary structure.

However, from the data collected from *circCHEK2* analysis (revised **Fig. 3,4,5**), editing-mediated structural changes of 6 additional editing-dependent ARcircs (revised **Fig. 4e**) and the RBP prediction (revised **Fig. 5h**), we hope to convince the reviewer that such editing-dependent mechanism can lead to changes in circRNA expression, either via altering the secondary structure or regulating the binding of RBPs or both.

Rebuttal Figure 2. a. Free energy change due to editing of RCMs. **≥3 AC**, RCMs containing ≥ 3 editing sites at A:C mismatches; **1 or 2 AC**, RCMs containing 1 or 2 editing sites at A:C mismatches; **non-AC**, RCMs do not contain editing sites at A:C mismatches.

b. Fold change in expression of circRNAs with ≥ 3 AC, 1 or 2 AC, or non-AC RCMs, from our circRNA-Seq data. Data are presented as box plots with median (horizontal line), 25–75 percentile (box), and 5–95 percentile (whisker) for each group and open dots indicate the outliers (Mann-Whitney test; ****, $p < 0.0001$; ns, not significant).

(7) *Fig 5b needs to be quantified with error bars.*

-As requested by the reviewer, we have quantified the relative band density of 3 independent experiments and presented in the revised **Fig. 5b**.

(8) *Fig 5c: The authors need to confirm that the expression of the linear RNA is not changed in these experiments.*

-Thank the reviewer for pointing it out. The expression of linear *CHEK2* transcript derived from the minigene remained unchanged upon overexpression of the edited minigene when compared to the wildtype minigene (revised **Supplementary Fig. 3b,c**), or upon overexpression of ADAR1 when compared to the EV control (revised **Supplementary Fig. 3d**).

(9) *P. 9: “Taken together, editing can not only alter the stability of secondary structure formed between RCMs, but also affect RBP binding to flanking intronic sequences, leading to changes in circRNA production.” This claim would be more justified if the authors could use CLIP and show that RBP binding to the intron is changed in cells, not just in vitro as was done in Fig 5b.*

-We thank the reviewer for giving this valuable suggestion. To further confirm that editing promotes the binding of PTBP1 on RCM *in vivo*, we performed PTBP1 RNA immunoprecipitation (RIP) in ADAR1-overexpressing or EV control cells followed by qPCR analysis using primers targeting the RCM region. We found that binding of PTBP1 to the *circCHEK2* RCM was significantly enhanced upon overexpression of ADAR1 (revised **Fig. 5c**).

Shen et al.
NCOMMS-21-13912-T

Intriguingly, the proportion of edited RCM transcripts (as reflected by editing frequencies of all 3 editing sites) was increased in PTBP1 RIP products when compared to the ‘Input’ samples, particularly in ADAR1-overexpressing cells (revised **Fig. 5d**), further confirming that PTBP1 preferentially binds to the edited *circCHEK2* RCM. In other words, editing can enhance PTBP1 binding to *circCHEK2* RCM region.

Minor points:

(1) Page 3: “*covalently closed continent loop structure*” should be “*covalently closed continuous loop structure*”
-Apologize for the typo. We have corrected it in the main text.

(2) Page 3: “*The first circRNA, which is a viroid in plants, was discovered and described in 1976.*” In the abstract, the authors have defined circRNAs as being produced by back-splicing and viroids are not made in this manner.
-We thank the reviewer for pointing it out. We have removed the description in the main text.

(3) Page 4: *The high expression of ADARs and circRNAs in the brain is a weak argument for why to study the connection between these molecules.*
-We thank the reviewer for pointing it out. We have removed the description in the main text.

(4) Page 10: Clarify in main text that a Cas13 method is being used.
-Thank the reviewer for the suggestion. We have included the information in the main text.

Reviewer #2:

The authors provided experimental evidence to support the hypothesis that A-to-I RNA editing can affect circRNA formation in a bidirectional manner, in which RNA editing alters the secondary structure formed between reverse complementary sequences residing in the introns flanking circularized exons through correcting A:C mismatches to I(G)-C pairs or creating I(G).U wobble pairs. They also found that RNA editing can affect RBP (e.g., PTBP1) recruitment and thereby regulate circRNA biogenesis. Finally, they showed that some ARcircs indeed influence tumorigenesis in multiple cell types. Their results modified the previous opinion that ADARs function as suppressor of circRNA formation by editing and melting the dsRNA. Some comments are listed below.

1. For identification of “high-confidence” A-to-I RNA editing events, the authors should evaluate the specificity of the identified sites by measuring (1) the fraction of A-to-G mismatches to all types of mismatches (%AG), (2) the ratio of #G-to-A mismatches to #A-to-G mismatches, and (3) the cis-preference of ADARs (the presence of “G” at the 5’ and 3’ neighbour nucleotides next to the A-to-I editing sites).

-Thank the reviewer for giving these suggestions. As requested, we measured the specificity of identified sites by calculating the proportion of each possible mismatches from our 3 RNA-Seq samples (empty vector, AR1 OE and AR2 OE). We assigned the mismatches based on the strand information from our stranded RNA-Seq datasets. As shown in **Rebuttal Fig. 3a**, A>G mismatches accounts for approximately 90% of all detected mismatches, consistent with previous studies reporting A-to-I editing as the most common type of RNA editing in humans (Athanasiadis et al., 2004; Li et al., 2009); while G>A type only account for 1.4%. Subsequently, only A>G sites with coverage ≥ 10 and absolute editing frequency > 0.1 were considered in the analysis.

To address the reviewer’s point (3), we extracted the sequence context of our high-confidence editing sites ($n = 41,551$) using “bedtools getfasta”, i.e. editing site plus 2 neighbour nucleotides on either side in a strand-specific manner. We then converted the nucleotide frequencies to a position probability matrix and plotted the sequence logo by using the “seqLogo” package. As shown in **Rebuttal Fig. 3b**, “G” is preferred to be excluded at 5’ neighbour while included at 3’ neighbour of editing sites, consistent with previous study (Wang et al., 2018). We have added these data into the revised **Supplementary Fig. 2a**, to support our identification of high-confidence editing sites.

Rebuttal Figure 3. a. Distribution of 12 types of mismatches from RNA sequencing data. b. Sequence preference of 2 neighbouring nucleotides surrounding A-to-I editing sites.

2. The following analyses should be included in the study: (1) the changes of editing levels after ADAR OE or ADAR KD; (2) the number of high-confidence A-to-I editing sites correcting A:C mismatches or creating wobble pairs within the RCMs regions; and (3) the expression levels of circRNAs affected by such an editing-dependent mechanism.

Shen et al.
NCOMMS-21-13912-T

-We thank the reviewer for the valuable suggestions. For the point (1), the change of editing levels after ADARs OE or KD has been published in our previous paper (Tang et al., 2020) which used the same RNA-Seq dataset. Please refer to the **Rebuttal Fig. 4**.

-To address the reviewer's point (2), we overexpressed the catalytic mutants of ADAR1 or ADAR2 (ADAR1-DeAD or ADAR2-DeAD) or the empty vector (EV) control in EC109 cells and performed circRNA-Seq to identify editing-dependent ARcircRNAs. First, in this new batch of circRNA-Seq, 76.3% (1,073/1,406) of ARcircRNAs identified from our previous circRNA-Seq could be detected again. Among these 1,073 ARcircRNAs, we defined editing-dependent ARcircRNAs using the following filter criteria: 1) there is no or minor change ($0.8 < \text{fold change} < 1.25$) in expression between the EV and DeAD-overexpressing cells; or 2) the pattern of change in expression upon overexpression of DeAD mutants is opposite to that of the wildtype ADAR1/2 when compared to the EV control. In total, we have obtained 767 editing-dependent circRNAs regulated by ADAR1 and/or ADAR2 (revised **Fig. 2e**). We previously examined the editing dependency for each of 20 candidate ARcircRNAs and identified 11 editing-dependent ARcircRNAs (**Fig. 2c,d**). Intriguingly, 9 out of 11 editing-dependent ARcircRNAs could be identified by our new circRNA-Seq analysis described above, indicating the reliability of our analysis of editing-dependent ARcircRNAs.

We then went on to identify editing sites within RCMs of these 767 editing-dependent ARcircRNAs. Since intronic regions tend to have low coverage in RNA-Seq datasets, when we applied the stringent criteria as described in **Fig. 2a** (total reads ≥ 10 in at least one sample and absolute editing frequency > 0.1), we obtained less than 100 editing sites located in RCMs of 47 editing-dependent ARcircRNAs, which are not sufficient for any following analysis. Therefore, to obtain more editing sites, we applied a less stringent filter criterion (≥ 1 variant read(s) supporting A>G mismatch in any sample) and identified 387 editing sites within RCMs of 89 editing-dependent ARcircRNAs. We then utilized RNAfold to computationally predict the secondary structure formed within RCMs of these editing-dependent ARcircRNAs. Editing sites detected in RCMs of 71 editing-dependent ARcircRNAs were predicted to reside in structured regions formed between RCMs, which allowed us to analyse the opposite nucleotides to the identified editing sites using a published script (Brummer et al., 2017). It is known that free energy of the structure is lowered, and thus, the stability of the structure is increased (Doshi et al., 2004; Wu et al., 2009). RNAfold, which can compute the minimum free energy (MFE), was used to calculate the change in MFE of the predicted secondary structure before and after editing. Correcting A:C mismatch to I:C pair leads to a more severe change on free energy and structure compared to altering A-U pair to I-U wobble pair (Vendeix et al., 2009; Wright et al., 2018). Our analysis confirmed that RCMs containing 1~2 or ≥ 3 editing sites at A:C mismatches (≥ 3 AC, $n=9$; 1 or 2 AC: $n=27$) tend to have a lower free energy post editing; while RCMs do not contain editing site(s) at A:C mismatches but with the majority at A-U pairs (non-AC, $n=35$) tend to have a higher free energy after editing (**Rebuttal Fig.**

2a). In sum, *in silico* analysis suggested that editing can alter the stability of dsRNA structures. To gain more experimental evidence, we generated RNA probes containing the wildtype (unedited) or edited (A-to-G mutations introduced at corresponding editing sites) RCM sequences for additional 6 editing-dependent ARcircRNAs (besides *circCHEK2*) and performed the native PAGE gel analysis. As expected, for 3 ADAR-promoted circRNAs (*circASH1L*, *circANKLE2-1* and *circRNF114*), the edited probes migrated faster in the gel than the wildtype probes (revised **Fig. 4c** and **Fig. 4e, left panel**); while for 3 ADAR-repressed circRNAs (*circSYNC*, *circDHX34* and *circRHOT1*), the edited probes migrated slightly slower (revised **Fig. 4d** and **Fig. 4e, right panel**). Altogether, we concluded that there could be a universal editing-dependent mechanism by which ADARs regulate circRNA biogenesis via editing-mediated change in the secondary structure formed by flanking introns.

-To address the reviewer's point (3), we went on to study the association between editing-mediated structural change and expression change of editing-dependent ARcircRNAs. Among these ARcircRNAs, we did not observe any significant change in expression between AC-containing or non-AC ARcircRNAs (**Rebuttal Fig. 2b**). In fact, it is not surprising to see such data, **for the following reasons**:

1) Editing may regulate circRNA biogenesis via changing the RBP binding sites and altering the binding affinity of RBPs to the target regions. In this study, we have shown that PTBP1 can preferentially bind to the edited RCM of *circCHEK2* (revised **Fig. 5**). In addition, as requested by reviewer 3, we conducted a large-scale analysis of RBP binding sites within the flanking introns of editing-dependent ARcircRNAs and showed that most RBPs included in the analysis have editing-mediated changes in their binding sites in flanking introns of more than 10 editing-dependent ARcircRNAs, even with the identified editing sites with stringent criteria (revised **Fig. 5h**).

2) Due to the technical limitations, a lot of intronic editing sites failed to be detected by RNA-Seq. In our study, despite using a less stringent filter, we only detected 387 editing sites in RCMs and thus identified a low number (71) of editing-dependent ARcircRNAs. This significantly affected our large-scale prediction of effects on circRNA levels based on the exact positions of editing sites and the localized secondary structure.

However, from the data collected from *circCHEK2* analysis (revised **Fig. 3,4,5**), editing-mediated structural changes of 6 additional editing-dependent ARcircRNAs (revised **Fig. 4e**) and the RBP prediction (revised **Fig. 5h**), we hope to convince the reviewer that such editing-dependent mechanism can lead to changes in circRNA expression, either via altering the secondary structure or regulating the binding of RBPs or both.

Rebuttal Figure 2. a. Free energy change due to editing of RCMs. **≥ 3 AC**, RCMs containing ≥ 3 editing sites at A:C mismatches; **1 or 2 AC**, RCMs containing 1 or 2 editing sites at A:C mismatches; **non-AC**, RCMs do not contain editing sites at A:C mismatches.

b. Fold change in expression of circRNAs with ≥ 3 AC, 1 or 2 AC, or non-AC RCMs, from our circRNA-Seq data. Data are presented as box plots with median (horizontal line), 25–75 percentile (box), and 5–95 percentile (whisker) for each group and open dots indicate the outliers (Mann-Whitney test; ****, $p < 0.0001$; ns, not significant).

3. The detailed information of the input sequence for RNA structure prediction should be provided. Did the input sequence include circularized exons?

-As the formation of secondary structure between flanking introns (RCMs) is one of the key factors manipulating circRNA biogenesis, the input sequence for RNA structure prediction includes RCM sequences without

Shen et al.
NCOMMS-21-13912-T

circularized exon(s). We have provided detailed information of the input sequences in the revised **Supplementary Table 5**.

4. For validations of the selected back-splicing events, all the Sanger sequencing results of the 21 cases should be provided (only 6 cases were illustrated in Supplementary Fig. 1c).

-Thank the reviewer for pointing it out. We have provided the Sanger sequencing results of 21 circRNAs in the revised **Supplementary Fig. 1d**.

5. Whether ADAR1 “indeed” binds and edits the dsRNA structure formed between RCMs located in the flanking introns of circCHEK2 should be experimentally confirmed.

-Thank the reviewer for the suggestion. We performed FLAG RNA immunoprecipitation (RIP) in EC109 cells transfected with FLAG-tagged ADAR1 (FLAG-ADAR1) or the empty vector control (FLAG-EV), followed by quantitative PCR (qPCR) using primers targeting RCM sequences within upstream and downstream flanking introns. As shown in the revised **Fig. 3b**, the RCM regions were significantly enriched in FLAG-RIP immunoprecipitates of FLAG-ADAR1 cells as compared to FLAG-EV cells (revised **Fig. 3b**), confirming the association of ADAR1 with the dsRNA structure formed between RCMs.

6. In Fig. 3b (pie charts), how did the authors get the editing levels of the three sites from endo RCM and minigene RCM? Were the editing levels validated by Sanger sequencing?

-Apologize for missing this important information. To reliably measure the editing frequency of each editing site, we first amplified the region containing 3 editing sites using PCR method. Purified PCR products were subcloned into the T-easy vector (Promega), and 20 to 22 individual plasmids were sequenced for each sample. For example, as shown in **Rebuttal Fig. 5**, a total of 20 plasmids from ADAR1 OE samples were sent for Sanger sequencing. We counted the number of unedited ‘A’ or edited ‘G’ clones and calculated the percentage of edited clones by $G/(A+G)$, i.e., $4/(16+4)=20.0\%$; **rebuttal Fig. 5**. The percentage of edited clones (a readout of ‘editing frequency’) was determined and shown by pie chart (represented by red slice). We have added the detailed information in the “Methods” and mentioned in the corresponding figure legend.

Rebuttal Figure 5. Sanger sequencing results of 20 plasmids generated by TA cloning showing the editing level of site #3 within RCM sequence of the endogenous *circCHEK2* transcript in ADAR1-overexpressing cells (Endo RCM, revised **Fig.3c**).

7. In figure 5b, the probe without RBP binding sites should be added as a negative control. In addition, to confirm the physical interactions between RBPs and the intron RCMs, the RNA immunoprecipitation experiments based on anti-RBPs should be performed. The binding affinity could be measured after ADAR OE or ADAR KD.

Shen et al.
NCOMMS-21-13912-T

-We thank the reviewer for giving these suggestions. We generated a mutant probe with all 12 predicted PTBP1 binding sites mutated as a negative control (NC) for RNA pulldown assay. As a result, PTBP1 was unable to bind with the mutant probe (revised **Supplementary Fig. 3a**).

As suggested by the reviewer, we performed PTBP1 RNA immunoprecipitation (RIP) in ADAR1-overexpressing or EV control cells followed by qPCR analysis using primers targeting the RCM region. We found that binding of PTBP1 to the *circCHEK2* RCM was significantly enhanced upon overexpression of ADAR1 (revised **Fig. 5c**). Intriguingly, the proportion of edited RCM transcripts (as reflected by editing frequencies of all 3 editing sites) was increased in PTBP1 RIP products when compared to the ‘Input’ samples, particularly in ADAR1-overexpressing cells (revised **Fig. 5d**), further confirming that PTBP1 preferentially binds to the edited *circCHEK2* RCM. In other words, editing can enhance PTBP1 binding to *circCHEK2* RCM region.

8. PTBP1 may affect ADAR1 translation (Cellular and Molecular Life Sciences 72.22 (2015): 4383-4397). Thus, it is important to explore the changes of ADAR1 protein expression after PTBP1 knockdown to rule out this possibility.

-We note the reviewer’s concern here. To address this, we checked the expression level of ADAR1 upon knockdown of PTBP1 in EC109 cells. However, we did not find significant reduction of ADAR1 protein level after silencing of PTBP1, possibly due to cell type specificity. We have included this data in revised **Supplementary Fig. 3e**.

9. In Fig. 6, it would be better to discuss why ADAR1 down-regulated circSLC39A8 exhibited the reduced tumorigenic ability upon circSLC39A8 knockdown. The difference of ADAR expression or editing capability between cancer and normal cells should be analyzed.

-We thank the reviewer for the suggestion. We have added the following information into the “Discussion”.
“The tumor promoting effect of ADAR1 is most likely arising from functional changes of multiple target genes via ADAR1’s editing-dependent and independent functions. The effect of ADAR1 or ADAR2 on tumorigenesis may differ depending on their target genes and/or cell or tissue types. For example, ADAR1-mediated protein-recoding editing of antizyme inhibitor 1 (*AZIN1*) promotes hepatocarcinogenesis (Chen et al., 2013); however, ADAR1-mediated editing can also suppress tumorigenesis by recoding Gamma-aminobutyric acid receptor subunit alpha-3 (*GABRA3*) in breast cancer (Tian et al., 2011). Likewise, each ARcirc may have its distinct cancer-related functions and thus, one should note that contribution of ADARs-regulated circRNA biogenesis to cancer is unlikely to be attributed to one single ARcirc.”

We have also included the data showing changes in expression of ADAR1 between HCC tumor and non-tumor tissues in the revised **Fig. 6e**.

10. Page 10, typo: “SNU198”  SNU398?

-Apologise for the typo. We have corrected it.

Shen et al.
NCOMMS-21-13912-T

Reviewer #3:

In this study, Shen et al. characterized ADARs as potent regulators of circular transcriptome through stabilizing or destabilizing secondary structures and/or changing the RBP binding. The authors further showed the functional effects of ARcircs in multiple cancers and highlighted the complexity of cross-talk in RNA processing and its contributions to tumorigenesis. The manuscript is overall interesting, significant, and well-organized. The reviewer has a few comments/suggestions:

1. It is interesting to show that A-to-I editing alters circRNA production via stabilizing or destabilizing dsRNA formed between RCMs (Fig. 4a). Could the authors apply similarly in silico secondary structure for all ARcircs to examine that if this can explain the distribution of promoted and suppressed circRNAs presented in fig. 1b?

-We thank the reviewer for the valuable suggestions. To address this point, we overexpressed the catalytic mutants of ADAR1 or ADAR2 (ADAR1-DeAD or ADAR2-DeAD) or the empty vector (EV) control in EC109 cells and performed circRNA-Seq to identify editing-dependent ARcircs. First, in this new batch of circRNA-Seq, 76.3% (1,073/1,406) of ARcircs identified from our previous circRNA-Seq could be detected again. Among these 1,073 ARcircs, we defined editing-dependent ARcircs using the following filter criteria: 1) there is no or minor change ($0.8 < \text{fold change} < 1.25$) in expression between the EV and DeAD-overexpressing cells; or 2) the pattern of change in expression upon overexpression of DeAD mutants is opposite to that of the wildtype ADAR1/2 when compared to the EV control. In total, we have obtained 767 editing-dependent circRNAs regulated by ADAR1 and/or ADAR2 (revised **Fig. 2e**). We previously examined the editing dependency for each of 20 candidate ARcircs and identified 11 editing-dependent ARcircs (**Fig. 2c,d**). Intriguingly, 9 out of 11 editing-dependent ARcircs could be identified by our new circRNA-Seq analysis described above, indicating the reliability of our analysis of editing-dependent ARcircs.

We then went on to identify editing sites within RCMs of these 767 editing-dependent ARcircs. Since intronic regions tend to have low coverage in RNA-Seq datasets, when we applied the stringent criteria as described in **Fig. 2a** (total reads ≥ 10 in at least one sample and absolute editing frequency > 0.1), we obtained less than 100 editing sites located in RCMs of 47 editing-dependent ARcircs, which are not sufficient for any following analysis. Therefore, to obtain more editing sites, we applied a less stringent filter criterion (≥ 1 variant read(s) supporting A>G mismatch in any sample) and identified 387 editing sites within RCMs of 89 editing-dependent ARcircs. We then utilized RNAfold to computationally predict the secondary structure formed within RCMs of these editing-dependent ARcircs. Editing sites detected in RCMs of 71 editing-dependent ARcircs were predicted to reside in structured regions formed between RCMs, which allowed us to analyse the opposite nucleotides to the identified editing sites using a published script (Brummer et al., 2017). It is known that free energy of the structure is lowered, and thus, the stability of the structure is increased (Doshi et al., 2004; Wu et al., 2009). RNAfold, which can compute the minimum free energy (MFE), was used to calculate the change in MFE of the predicted secondary structure before and after editing. Correcting A:C mismatch to I-C pair leads to a more severe change on free energy and structure compared to altering A-U pair to I-U wobble pair (Vendeix et al., 2009; Wright et al., 2018). Our analysis confirmed that RCMs containing 1~2 or ≥ 3 editing sites at A:C mismatches (≥ 3 AC, $n=9$; 1 or 2 AC: $n=27$) tend to have a lower free energy post editing; while RCMs do not contain editing site(s) at A:C mismatches but with the majority at A-U pairs (non-AC, $n=35$) tend to have a higher free energy after editing (**Rebuttal Fig. 2a**). In sum, *in silico* analysis suggested that editing can alter the stability of dsRNA structures. To gain more experimental evidence, we generated RNA probes containing the wildtype (unedited) or edited (A-to-G mutations introduced at corresponding editing sites) RCM sequences for additional 6 editing-dependent ARcircs (besides *circCHEK2*) and performed the native PAGE gel analysis. As expected, for 3 ADAR-promoted circRNAs (*circASH1L*, *circANKLE2-1* and *circRNF114*), the edited probes migrated faster in the gel than the wildtype probes (revised **Fig. 4c** and **Fig. 4e, left panel**); while for 3 ADAR-repressed circRNAs (*circSYNC*, *circDHX34* and *circRHOT1*), the edited probes migrated slightly slower (revised **Fig. 4d** and **Fig. 4e, right panel**). Altogether, we concluded that there could be a universal editing-dependent mechanism by which ADARs regulate circRNA biogenesis via editing-mediated change in the secondary structure formed by flanking introns.

-We then went on to study the association between editing-mediated structural change and expression change of editing-dependent ARcircs. Among these ARcircs, we did not observe any significant change in

expression between AC-containing or non-AC ARcircs (**Rebuttal Fig. 2b**). In fact, it is not surprising to see such data, **for the following reasons**:

1) Editing may regulate circRNA biogenesis via changing the RBP binding sites and altering the binding affinity of RBPs to the target regions. In this study, we have shown that PTBP1 can preferentially bind to the edited RCM of *circCHEK2* (revised **Fig. 5**). In addition, as requested in comment #3, we conducted a large-scale analysis of RBP binding sites within the flanking introns of editing-dependent ARcircs and showed that most RBPs included in the analysis have editing-mediated changes in their binding sites in flanking introns of more than 10 editing-dependent ARcircs, even with the identified editing sites with stringent criteria (revised **Fig. 5h**).

2) Due to the technical limitations, a lot of intronic editing sites failed to be detected by RNA-Seq. In our study, despite using a less stringent filter, we only detected 387 editing sites in RCMs and thus identified a low number (71) of editing-dependent ARcircs. This significantly affected our large-scale prediction of effects on circRNA levels based on the exact positions of editing sites and the localized secondary structure.

However, from the data collected from *circCHEK2* analysis (revised **Fig. 3,4,5**), editing-mediated structural changes of 6 additional editing-dependent ARcircs (revised **Fig. 4e**) and the RBP prediction (revised **Fig. 5h**), we hope to convince the reviewer that such editing-dependent mechanism can lead to changes in circRNA expression, either via altering the secondary structure or regulating the binding of RBPs or both.

Rebuttal Figure 2. a. Free energy change due to editing of RCMs. ≥ 3 AC, RCMs containing ≥ 3 editing sites at A:C mismatches; 1 or 2 AC, RCMs containing 1 or 2 editing sites at A:C mismatches; non-AC, RCMs do not contain editing sites at A:C mismatches.

b. Fold change in expression of circRNAs with ≥ 3 AC, 1 or 2 AC, or non-AC RCMs, from our circRNA-Seq data. Data are presented as box plots with median (horizontal line), 25–75 percentile (box), and 5–95 percentile (whisker) for each group and open dots indicate the outliers (Mann-Whitney test; ****, $p < 0.0001$; ns, not significant).

2. It is significant to conclude that ADARs-mediated circRNA regulation exists in multiple cancer types. The authors showed very nice experimental data in multiple cancer cell lines. Is it possible for the authors to confirm this pattern in patient data (author's own data or public data) – for example, compare the ARcircs expression in ADAR high vs. ADAR low expression samples.

-Thank the reviewer for the suggestion. In the revised **Fig. 6e, f**, we investigated 1) the expression pattern of *circCHEK2* and 2) the association between expression levels of *ADAR1* and *circCHEK2* in 17 matched pairs of primary HCC and non-tumor (NT) liver samples as well as 20 matched pairs of primary colorectal cancer (CRC) and NT colon samples. We found that 41% (7 out of 17) and 60% (12 out of 20) of HCC and CRC patients demonstrated a ≥ 2 -fold increase in *circCHEK2* expression in tumors compared to their NT samples, respectively (**Fig. 6e,f**, upper panels). Next, both HCC and CRC patients were stratified into 2 groups: ADAR1-down and ADAR1-up, based on the decreased or increased expression of ADAR1 in tumors compared to their matched NT samples, respectively (**Fig. 6e,f**, lower panels). We found that in the ADAR1-down or ADAR1-up group of HCC patients, 3 out of 6 (50%) or 6 out of 11 (54.5%) showed ≥ 2 -fold decrease or increase in *circCHEK2* expression in tumors, respectively (**Fig. 6e**). Likewise, in the ADAR1-down or ADAR1-up group of CRC patients, 4 out of 6 (67%) or 8 out of 14 (57%) showed ≥ 2 -fold decrease or increase in *circCHEK2* expression in tumors, respectively

Shen et al.
NCOMMS-21-13912-T

(Fig. 6f). These findings suggested that ADARs-mediated circRNA regulation is most likely present in multiple cancer types.

3. It is interesting to show that two RBPs with a respective binding motif near circCHEK2. How about other RBPs for other ARcircs? Since recent studies showed that other RBPs may also regulate the circRNA expression (e.g., PMID: 31446897). A computational analysis regarding other RBPs/circRNAs will enhance the conclusion.

-We thank the reviewer for the suggestion. We retrieved the sequences surrounding 571 high-confidence editing sites (± 10 nt) within the flanking introns (not limited to RCM sequences) of 92 editing-dependent ARcircs and analysed RBP binding motifs before and after editing using RBPmap (Paz et al., 2014). We then calculated the number of circRNAs which have altered RBP binding sites on flanking introns due to editing. We found that among 132 analysed RBPs with annotated binding sites in RBPmap, 129 RBPs, including PTBP1 and those which have been shown to regulate circRNA biogenesis such as MBNL1 (Ashwal-Fluss et al., 2014), FUS (Errichelli et al., 2017), SFPQ (Stagsted et al., 2021), HNRNPL (Fei et al., 2017), KHSRP (Fei et al., 2017), and QKI (Conn et al., 2015), were found to have editing-mediated changes in their binding sites at flanking introns of more than 10 editing-dependent ARcircs (revised **Fig. 5h** and **Supplementary Table 4**), implying that altering RBP binding affinity is an important mechanism for editing to regulate circRNA biogenesis.

4. The authors mentioned the controversial results from previous studies (references 19, 20, 24) in the discussion. Could the authors explain the different observations?

- In one of previous studies, the authors identified circular RNAs by performing the total RNA-Seq (Ivanov et al., 2015) ^{ref.19}. Total RNA-Seq is not an optimal method to detect circular RNAs as most detected transcripts are linear RNAs (Jeck et al., 2013). The other 2 studies only showed several circRNAs which were regulated by ADAR1 without further transcriptome-wide investigation of circRNA regulation by ADAR proteins (Aktas et al., 2017; Shi et al., 2017) ^{refs.20, 24}. In our study, we applied RNase R treatment prior to RNA sequencing which could enrich circRNAs (Jeck et al., 2013), so that we could identify thousands of high-confidence ARcircs and reveal the bidirectional regulation of ADARs on circRNAs.

5. GSE164681 is not accessible.

- Please access at <https://www.ncbi.nlm.nih.gov/geo/query/acc.cgi?acc=GSE164681> with token: azwbsigyvbcrdin.

References

- Aktas, T., Avsar Ilik, I., Maticzka, D., Bhardwaj, V., Pessoa Rodrigues, C., Mittler, G., Manke, T., Backofen, R., and Akhtar, A. (2017). DHX9 suppresses RNA processing defects originating from the Alu invasion of the human genome. *Nature* *544*, 115-119.
- Ashwal-Fluss, R., Meyer, M., Pamudurti, N.R., Ivanov, A., Bartok, O., Hanan, M., Evtantal, N., Memczak, S., Rajewsky, N., and Kadener, S. (2014). circRNA biogenesis competes with pre-mRNA splicing. *Mol Cell* *56*, 55-66.
- Athanasiadis, A., Rich, A., and Maas, S. (2004). Widespread A-to-I RNA editing of Alu-containing mRNAs in the human transcriptome. *PLoS Biol* *2*, e391.
- Brummer, A., Yang, Y., Chan, T.W., and Xiao, X. (2017). Structure-mediated modulation of mRNA abundance by A-to-I editing. *Nat Commun* *8*, 1255.
- Chen, L., Li, Y., Lin, C.H., Chan, T.H., Chow, R.K., Song, Y., Liu, M., Yuan, Y.F., Fu, L., Kong, K.L., *et al.* (2013). Recoding RNA editing of AZIN1 predisposes to hepatocellular carcinoma. *Nat Med* *19*, 209-216.
- Chen, L., Wang, C., Sun, H., Wang, J., Liang, Y., Wang, Y., and Wong, G. (2021). The bioinformatics toolbox for circRNA discovery and analysis. *Brief Bioinform* *22*, 1706-1728.
- Conn, S.J., Pillman, K.A., Toubia, J., Conn, V.M., Salmanidis, M., Phillips, C.A., Roslan, S., Schreiber, A.W., Gregory, P.A., and Goodall, G.J. (2015). The RNA binding protein quaking regulates formation of circRNAs. *Cell* *160*, 1125-1134.

Shen et al.
NCOMMS-21-13912-T

Doshi, K.J., Cannone, J.J., Cobaugh, C.W., and Gutell, R.R. (2004). Evaluation of the suitability of free-energy minimization using nearest-neighbor energy parameters for RNA secondary structure prediction. *BMC Bioinformatics* 5, 105.

Errichelli, L., Dini Modigliani, S., Laneve, P., Colantoni, A., Legnini, I., Capauto, D., Rosa, A., De Santis, R., Scarfò, R., Peruzzi, G., *et al.* (2017). FUS affects circular RNA expression in murine embryonic stem cell-derived motor neurons. *Nat Commun* 8, 14741.

Fei, T., Chen, Y., Xiao, T., Li, W., Cato, L., Zhang, P., Cotter, M.B., Bowden, M., Lis, R.T., Zhao, S.G., *et al.* (2017). Genome-wide CRISPR screen identifies HNRNPL as a prostate cancer dependency regulating RNA splicing. *Proc Natl Acad Sci U S A* 114, E5207-e5215.

Gao, Y., Zhang, J., and Zhao, F. (2018). Circular RNA identification based on multiple seed matching. *Brief Bioinform* 19, 803-810.

Ivanov, A., Memczak, S., Wyler, E., Torti, F., Porath, H.T., Orejuela, M.R., Piechotta, M., Levanon, E.Y., Landthaler, M., Dieterich, C., *et al.* (2015). Analysis of intron sequences reveals hallmarks of circular RNA biogenesis in animals. *Cell Rep* 10, 170-177.

Jeck, W.R., Sorrentino, J.A., Wang, K., Slevin, M.K., Burd, C.E., Liu, J., Marzluff, W.F., and Sharpless, N.E. (2013). Circular RNAs are abundant, conserved, and associated with ALU repeats. *RNA (New York, N.Y.)* 19, 141-157.

Li, J.B., Levanon, E.Y., Yoon, J.K., Aach, J., Xie, B., Leproust, E., Zhang, K., Gao, Y., and Church, G.M. (2009). Genome-wide identification of human RNA editing sites by parallel DNA capturing and sequencing. *Science* 324, 1210-1213.

Paz, I., Kostı, I., Ares, M., Jr., Cline, M., and Mandel-Gutfreund, Y. (2014). RBPmap: a web server for mapping binding sites of RNA-binding proteins. *Nucleic Acids Res* 42, W361-367.

Shi, L., Yan, P., Liang, Y., Sun, Y., Shen, J., Zhou, S., Lin, H., Liang, X., and Cai, X. (2017). Circular RNA expression is suppressed by androgen receptor (AR)-regulated adenosine deaminase that acts on RNA (ADAR1) in human hepatocellular carcinoma. *Cell Death Dis* 8, e3171.

Stagsted, L.V.W., O'Leary, E.T., Ebbesen, K.K., and Hansen, T.B. (2021). The RNA-binding protein SFPQ preserves long-intron splicing and regulates circRNA biogenesis in mammals. *Elife* 10, e63088.

Tang, S.J., Shen, H., An, O., Hong, H., Li, J., Song, Y., Han, J., Tay, D.J.T., Ng, V.H.E., Bellido Molias, F., *et al.* (2020). Cis- and trans-regulations of pre-mRNA splicing by RNA editing enzymes influence cancer development. *Nat Commun* 11, 799.

Tian, N., Yang, Y., Sachsenmaier, N., Muggenheimer, D., Bi, J., Waldsich, C., Jantsch, M.F., and Jin, Y. (2011). A structural determinant required for RNA editing. *Nucleic Acids Res* 39, 5669-5681.

Vendeix, F.A., Munoz, A.M., and Agris, P.F. (2009). Free energy calculation of modified base-pair formation in explicit solvent: A predictive model. *RNA (New York, N.Y.)* 15, 2278-2287.

Wang, Y., Park, S., and Beal, P.A. (2018). Selective Recognition of RNA Substrates by ADAR Deaminase Domains. *Biochemistry* 57, 1640-1651.

Wright, D.J., Force, C.R., and Znosko, B.M. (2018). Stability of RNA duplexes containing inosine-cytosine pairs. *Nucleic Acids Res* 46, 12099-12108.

Wu, J.C., Gardner, D.P., Ozer, S., Gutell, R.R., and Ren, P. (2009). Correlation of RNA secondary structure statistics with thermodynamic stability and applications to folding. *J Mol Biol* 391, 769-783.

Zhang, X.O., Dong, R., Zhang, Y., Zhang, J.L., Luo, Z., Zhang, J., Chen, L.L., and Yang, L. (2016). Diverse alternative back-splicing and alternative splicing landscape of circular RNAs. *Genome Res* 26, 1277-1287.

REVIEWERS' COMMENTS

Reviewer #1 (Remarks to the Author):

The authors have adequately addressed my prior concerns.

I have very two minor suggestions:

(1) Line 126: The stringent filter criteria do not address whether the ARcircs in particular were also predicted by CIRI2 or CIRCexplorer2. Please provide this information in the manuscript. As mentioned in my prior review, circRNAs predicted by a single algorithm are often of unclear significance so this easy to produce information is helpful for giving a reader more perspective.

(2) Figure 7a: The abbreviations “gNT-circular” etc are a bit non-intuitive as they describe both the guide RNA sequence and the RNA being examined by qPCR. Please add an additional sentence in the legend to help less experienced readers understand the notation.

Reviewer #2 (Remarks to the Author):

I am glad to see the revision, which has addressed all my concerns. The authors have done a good job for the revision. I am just curious about Rebuttal Figure 4. As far as I know, editing levels are generally much lower than 10% in human. We can find that the median values of editing levels are greater than 30% before ADAR1/ADAR2 knockdown (Rebuttal Figure 4, top), while the median values are only ~30% after ADAR1/ADAR2 overexpression (Rebuttal Figure 4, bottom). The median values remain greater than 10% after the knockdown (Rebuttal Figure 4, top).

Reviewer #3 (Remarks to the Author):

The authors revised the manuscript accordingly, and I do not have any further comments.

Reviewer #1 (Remarks to the Author):

The authors have adequately addressed my prior concerns.

I have very two minor suggestions:

(1) Line 126: The stringent filter criteria do not address whether the ARcircs in particular were also predicted by CIRI2 or CIRCexplorer2. Please provide this information in the manuscript. As mentioned in my prior review, circRNAs predicted by a single algorithm are often of unclear significance so this easy to produce information is helpful for giving a reader more perspective.

-- As suggested by the reviewer, we applied the same stringent filter criteria to identify ADARs-regulated circRNAs using CIRCexplorer2. Overall, approximately 93% (1,313/1,406) of ARcircs identified by our in-house pipeline could be detected by CIRCexplorer2 and demonstrates the same pattern of changes upon modulation of ADAR1/2 expression. We have included the data in the revised Supplementary Figure 1b and provided the full list of ARcircs identified by CIRCexplorer2 in Supplementary Data 1.

(2) Figure 7a: The abbreviations “gNT-circular” etc are a bit non-intuitive as they describe both the guide RNA sequence and the RNA being examined by qPCR. Please add an additional sentence in the legend to help less experienced readers understand the notation.

--Thank the reviewer for the suggestion. We have provided additional information (shown below) to the figure legend.

“gNT-circular, circRNA expression of indicated gene upon treatment of CasRX and non-targeting guide RNA, and so forth.”

Reviewer #2:

I am glad to see the revision, which has addressed all my concerns. The authors have done a good job for the revision. I am just curious about Rebuttal Figure 4. As far as I know, editing levels are generally much lower than 10% in human. We can find that the median values of editing levels are greater than 30% before ADAR1/ADAR2 knockdown (Rebuttal Figure 4, top), while the median values are only ~30% after ADAR1/ADAR2 overexpression (Rebuttal Figure 4, bottom). The median values remain greater than 10% after the knockdown (Rebuttal Figure 4, top).

--We thank the reviewer for raising this question. As reported by our group and others previously^{1,2}, ESCC tumors were found to have a higher level of A-to-I editing compared to non-tumor samples. EC109 is an ESCC cell line which has been widely used for RNA editing research^{1,3,4}. It is therefore not surprising to observe a higher global editing level in EC109 cells. Moreover, in our study, the ADAR1/2-overexpressing or knockdown EC109 cells were established using different lentiviral expression systems (pLenti6 for overexpression while pLKO.1 for knockdown), and these cells were selected using different antibiotics. In addition, when we performed transcriptome-wide RNA editing analysis in previous Rebuttal Figure 4, we viewed these samples as two independent groups of cells (**KD** and **OE** groups). Only those editing sites with ≥ 20 total reads in all samples within the same group (e.g., **KD** group: shScr, shADAR1#3, and shADAR#9) were included for the analysis. Therefore, we did not expect to obtain the totally same set of editing sites between KD and OE groups. With the clarifications above, we hope to convince the reviewer that it may not be fair to compare

the global editing level between two different groups of cells.

Reviewer #3:

The authors revised the manuscript accordingly, and I do not have any further comments.

References

1. Qin, Y.R. et al. Adenosine-to-inosine RNA editing mediated by ADARs in esophageal squamous cell carcinoma. *Cancer Res* **74**, 840–51 (2014).
2. Paz-Yaacov, N. et al. Elevated RNA Editing Activity Is a Major Contributor to Transcriptomic Diversity in Tumors. *Cell Rep* **13**, 267–76 (2015).
3. Hong, H. et al. Bidirectional regulation of adenosine-to-inosine (A-to-I) RNA editing by DEAH box helicase 9 (DHX9) in cancer. *Nucleic Acids Res* **46**, 7953–7969 (2018).
4. Tang, S.J. et al. Cis- and trans-regulations of pre-mRNA splicing by RNA editing enzymes influence cancer development. *Nat Commun* **11**, 799 (2020).